# Modeling the Diurnal Variability of Agricultural Ammonia in Bakersfield, California during the CalNex Campaign

C. R. Lonsdale[1], J. D. Hegarty[1], K. E. Cady-Pereira[1], M. J. Alvarado[1], D. K. Henze[2], M. D. Turner[2], S. L. Capps[2], J. B. Nowak[3,4,*], J. A. Neuman[4], A. M. Middlebrook[4], R. Bahreini[3,4,**], J. G. Murphy[5], M. Z. Markovic[5,***], T. C. VandenBoer[5,****], L. M. Russell[6], A. J. Scarino[7]

[1]Atmospheric and Environmental Research, Lexington, MA, USA
[2]Department of Mechanical Engineering, University of Colorado, Boulder, CO, USA
[3]Cooperative Institute for Research in Environmental Sciences, University of Colorado Boulder, CO, USA
[4]Chemical Sciences Division, Earth System Research Lab, NOAA, Boulder, CO, USA
[5]Department of Chemistry, University of Toronto, Toronto, ON Canada
[6]Scripps Institution of Oceanography, University of California, San Diego, CA, USA
[7]Science Systems and Applications Inc., Hampton, VA, USA
*Now at Aerodyne Research, Inc. Billerica, MA, USA
** Now at Department of Environmental Sciences, University of California, Riverside, CA, USA
***Now at Picarro, Inc., Santa Clara, CA, USA
**** Now at Department of Earth Sciences, Memorial University of Newfoundland, NL Canada

*Correspondence to*: C. R. Lonsdale (clonsdal@aer.com)

**Abstract.** $NH_3$ retrievals from the NASA Tropospheric Emission Spectrometer (TES), as well as surface and aircraft observations of $NH_{3(g)}$ and submicron $NH_{4(p)}$, are used to evaluate modeled concentrations of $NH_{3(g)}$ and $NH_{4(p)}$ from the Community Multiscale Air Quality (CMAQ) model in the San Joaquin Valley (SJV) during the California Research at the Nexus of Air Quality and Climate Change (CalNex) campaign. We find that simulations of $NH_3$ driven with the California Air Resources Board (CARB) emission inventory are qualitatively and spatially consistent with TES satellite observations, with a correlation coefficient ($r^2$) of 0.64. However, the surface observations at Bakersfield indicate a diurnal cycle in the model bias, with CMAQ overestimating surface $NH_3$ at night and underestimating it during the day. The surface, satellite, and aircraft observations all suggest that daytime $NH_3$ emissions in the CARB inventory are underestimated by at least a factor of two, while the nighttime overestimate of $NH_{3(g)}$ is likely due to a combination of overestimated $NH_3$ emissions and underestimated deposition.

Running CMAQ v5.0.2 with the bi-directional $NH_3$ scheme reduces $NH_3$ concentrations at night and increases them during the day. This reduces the model bias when compared to the surface and satellite observations, but the increased concentrations aloft significantly increase the bias relative to the aircraft observations. We attempt to further reduce model bias by using the surface observations at Bakersfield to derive an empirical diurnal cycle of $NH_3$ emissions in the SJV, in which nighttime and midday emissions differ by about a factor of 4.5. Running CMAQv5.0.2 with a bi-directional $NH_3$ scheme together with this emissions diurnal profile further reduces model bias relative to the surface observations.

Comparison of these simulations with the vertical profile retrieved by TES shows little bias except for the lowest retrieved level, but the model bias relative to flight data aloft increases slightly. Our results indicate that both diurnally-varying emissions and a bi-directional $NH_3$ scheme should be applied when modeling $NH_{3(g)}$ and $NH_{4(p)}$ in this region. The remaining model errors suggest that the bi-directional $NH_3$ scheme in CMAQ v5.0.2 needs further improvements to shift the peak $NH_3$ land-atmosphere flux to earlier in the day. We recommend that future work include: updates to the current CARB $NH_3$ inventory to account for $NH_3$ from fertilizer application, livestock, and other farming practices separately; adding revised information on crop management practices specific to the SJV region to the bi-directional $NH_3$ scheme; and top-down studies focused on determining the diurnally-varying biases in the canopy compensation point that determines the net land-atmosphere $NH_3$ fluxes.

**1 Introduction**

The emissions of ammonia ($NH_3$) to the atmosphere are highly uncertain (e.g., Pinder et al., 2006; Beusen et al., 2008; Galloway et al., 2008; Henze et al., 2009; Schlesinger, 2009). Nitrogen dioxide ($NO_x = NO + NO_2$) and sulfur dioxide ($SO_2$) photo-oxidize in the atmosphere to form nitric acid ($HNO_3$) and sulfuric acid ($H_2SO_4$), respectively, which react with atmospheric gas-phase ammonia ($NH_{3(g)}$) to form ammonium sulfate (($NH_4$)$_2SO_4$) and ammonium nitrate ($NH_4NO_3$) aerosol. Uncertainty in $NH_3$ emissions therefore leads to significant uncertainties in the concentrations of secondary inorganic aerosols. Ammonium sulfate and nitrate aerosols contribute to fine particulate matter concentrations ($PM_{2.5}$), and thus to decreased visibility, altered climate, and acidification and eutrophication in sensitive ecosystems (e.g., Paulot et al., 2014; RoTAP. 2012; Bricker et al., 2007; Martin et al., 2004).

$PM_{2.5}$ also causes adverse health effects (WHO, 2016; Pope et al., 2004). In particular, some regions in the San Joaquin Valley (SJV) in California have been designated as non-attainment areas for $PM_{2.5}$, with $NH_3$ emissions contributing to more than half of the inorganic $PM_{2.5}$ in the state (Schiferl et al., 2014), depending on ambient conditions and concentrations (Lonsdale et al., 2012). During the NOAA California Research at the Nexus of Air Quality and Climate Change (CalNex) campaign in May and June of 2010, however, concentrations of $PM_{2.5}$ rarely exceeded the National Ambient Air Quality Standard (NAAQS) in the SJV, as $PM_{2.5}$ exceedances here generally occur in the winter. While emissions of $NO_x$ and $SO_2$ are relatively well constrained, are regulated by the United States Environmental Protection Agency (US EPA), and are predicted to continually decrease due to air quality regulations and emission reducing technologies (US EPA, 2010), $NH_3$ emissions are not currently regulated and are predicted to stay constant or increase in the US over the next several decades in the US due to an increasing population and associated increases in farming and agricultural activities (Moss et al., 2010). Climate change is also predicted to increase $NH_3$ emissions (+0-40 % in north-central Europe) with larger countries having the largest uncertainty in emissions variations (Skjøth et al., 2013).

Anthropogenic $NH_3$ sources in the SJV are dominated by agricultural activities, with livestock waste estimated to contribute about 74 % of total anthropogenic $NH_3$ to the atmosphere and chemical fertilizer use another 16 % (Simon et al., 2008).

Agricultural emissions of $NH_3$ can be highly variable due to factors such as the differences in fertilizer application, the diet provided to livestock, and waste management and storage practices of farmers (Hristov et al., 2011; Sawycky et al., 2014). In addition, while $NH_{3(g)}$ can be quickly deposited to the surface causing soil acidification, water eutrophication, and an imbalance of ecosystems when in excess (e.g., Carfrae et al., 2004), the air-surface exchange of $NH_3$ is bi-directional, with the direction of the $NH_3$ flux between the land and the atmosphere varying with temperature, relative humidity, vegetation and soil type, maintenance (e.g., cutting and tilling practices), and fertilizer applications (Nemitz et al., 2001; Zhang et al., 2010; Ellis et al., 2011; Bash et al., 2013; Sawycky et al., 2014). This complexity in the emission and deposition of $NH_3$, along with the rapid reactions of $NH_3$ with $HNO_3$ and $H_2SO_4$ and the consequently short (~1 day) atmospheric lifetime of $NH_3$, leads to large temporal and spatial variability as seen in in situ measurements (e.g., Langford et al., 1992; Carmichael et al., 2003; Nowak et al., 2010; Walker et al., 2013) and in satellite retrievals (e.g., Clarisse et al., 2013; Pinder et al., 2011; Shephard et al., 2011; Heald et al., 2012; Sun et al., 2015; Shephard and Cady-Pereira, 2015; Shephard et al., 2015).

Recent studies have recognized a diurnal pattern in $NH_3$ emissions from livestock attributed to potential differences in farm management practices, livestock housing outflow patterns, and variations in soil moisture, temperature, and wind speed (Hensen et al., 2009; Zhu et al., 2015a; Zhu et al., 2015b). To account for this, a diurnal variability scheme was implemented into global simulations using the global 3-dimensional chemical transport model, GEOS-Chem, and was shown to decrease $NH_3$ concentrations globally (Zhu et al., 2015a). That study also calculated the bi-directional exchange of $NH_3$, which decreased $NH_3$ concentrations in the US in the months of October through April and increased it in the month of July (Zhu et al., 2015a). Bash et al. (2013) also explored the sensitivity of modeled $NH_3$ concentrations to a bi-directional $NH_3$ scheme that used meteorological factors, including temperature, wind speed, agricultural crop flux values, and a nitrogen soil geochemistry parameterization in the CMAQ model. They found that over the continental US their model run with the bi-directional $NH_3$ scheme decreased the total dry deposition of $NH_3$ by 45 %, thus increasing atmospheric $NH_3$ concentrations and $NH_x$ wet deposition by 10 % and 14 %, respectively. Wichink Kruit et al. (2012) use the DEPosition of Acidifying Compounds (DEPAC) surface-atmospheric exchange module in a CTM and saw an increase in atmospheric $NH_3$ almost everywhere in their model domain, including decreased $NH_3$ deposition with a remaining underestimation in agricultural areas.

Previous studies have also shown that errors in $NH_3$ emissions are a common contributing factor to modeled $PM_{2.5}$ and $NH_3$ bias (e.g., Schiferl et al., 2014). Skjøth et al., (2011) discuss their method for calculating dynamic $NH_3$ emissions that includes distributions of agricultural $NH_3$ in Europe. Their method is designed for use in chemical transport models and their results show considerable improvements made in the agricultural $NH_3$ sector, particularly in areas with detailed records of agricultural practices. Inverse modeling studies have been used to reduce the uncertainty in $NH_3$ emissions as well, generally by assimilating surface observations of the wet deposition of ammonium ($NH_4^+$) in precipitation. Gilliland et al. (2003) used the CMAQ model to determine that the 1990 version of the US EPA National Emissions Inventory (NEI) overestimated total emissions of $NH_3$ by 20 %. Gilliland et al. (2006) performed a similar study for the 2001 NEI and found that total emissions of $NH_3$ were represented well, but needed to be increased in summer and reduced in winter. Henze et al. (2009) used the

adjoint of the global chemical transport model GEOS-Chem to assimilate the Inter Agency Monitoring of Protected Visual Environments (IMPROVE) observations and found that total US $NH_3$ emissions for 1998 were overestimated.

More recently, satellite observations of $NH_3$ have been incorporated into inverse studies. By assimilating satellite retrievals of $NH_3$ concentrations from the Tropospheric Emission Spectrometer (TES) (Beer et al., 2008; Shephard et al., 2011) aboard the NASA Aura satellite, it has been found that $NH_3$ emission sources in GEOS-Chem are broadly underestimated (Zhu et al., 2013). Heald et al. (2012) and Walker et al. (2012) used IMPROVE data and satellite retrievals of $NH_3$ from the Infrared Atmospheric Sounding Instrument (IASI, Van Damme et al., 2014) to show that $NH_3$ emissions are likely underestimated in GEOS-Chem for California, leading to a local underestimate of $NH_{4(p)}$. Other infrared nadir sounders have been used to provide satellite observations of $NH_3$. For example, Shephard and Cady-Pereira (2015) demonstrated the ability of the Crosstrack Infrared Sounder (CrIS) aboard the joint NOAA-NASA Suomi National Polar-orbiting satellite to measure daily, spatially distributed tropospheric $NH_3$ in California, and in preliminary results found it correlated well with Deriving Information on Surface Conditions from Column and Vertically Resolved Observations Relevant to Air Quality (DISCOVER-AQ) aircraft measurements in the SJV in January 2013.

Investigating the formation, transport, and fate of $NH_{3(g)}$ and $NH_{4(p)}$ in California was one of the major goals of the CalNex field campaign, which provided measurements from flights and surface sites (Ryerson et al., 2012) in the Los Angeles basin and in the Central Valley. Nowak et al. (2012) used these data to demonstrate the importance of ammonium nitrate formation downwind of the Los Angeles urban core and dairy facilities further east. They found that $NH_3$ emissions from these dairy farms were underestimated by a factor of 3 or more, thus indicating the need for better representation in this emission sector. Kelly et al. (2014) in general saw well-correlated comparisons of CMAQ model estimates to measurements from the EPA's Chemical Speciation Network. Their model tended to under-predict $NH_x$ ($NH_x = NH_{3(g)} + NH_{4(p)}$) during the day at the Bakersfield, CA site and significantly over-predict $NH_{3(g)}$ at night. They suggest that this model bias may be due to emissions from livestock and dairy farms being too low and lacking in variability in this region or to errors in crustal cation predictions and the missing effects of organic acids and amines on inorganic aerosol thermodynamics (Kelly et al., 2014).

Model estimates of the planetary boundary layer (PBL) height are essential in correctly quantifying changes in atmospheric pollutant concentrations, especially for short-lived pollutants like $NH_3$. Such estimates are difficult at fine spatial and temporal scales, especially in the complex terrain of the SJV. Scarino et al. (2014) studied the PBL and mixed layer heights during CalNex using WRF and high spectral resolution lidar (HSRL) data taken during the campaign. They found that, in general, there is good agreement between the WRF modeled output and measured values; however, in the California Central Valley there is a WRF mixed-layer height over-prediction and an inability to represent the diurnal growth of the mixed layer in the early part of the day. Additionally they suggest that future improvements will require a focus on mixing layer characteristics, soil moisture, and temperature. Baker et al. (2013) explored how well the WRF model configuration used to drive the CMAQ simulations of Kelly et al. (2014) simulates PBL height during CalNex, using two versions of WRF. The study shows that both WRF versions simulate the PBL and mixing layers well within the SJV, as well as other large scale flow patterns, but under-predict local wind speed and temperature. A strong aerosol gradient is used to identify the top of the

PBL in HSRL measurements; this strong gradient may also be present in a nighttime residual layer. Baker et al. (2013) take this into account by identifying the surface-attached mixed layer, which they assume as the lowest significant gradient in such a circumstance.

In this study, we use the CalNex observations of $NH_{3(g)}$ and $NH_{4(p)}$ and the CMAQ model to evaluate the estimates of $NH_3$
emissions in the SJV contained in the California Air Resources Board (CARB) inventory (Figure 1). While previous $NH_3$ model evaluation efforts using CalNex data have focused on the NEI inventory (Kelly et al., 2014; Heald et al., 2012; Walker et al., 2012), the CARB inventory is used in the development of California's State Implementation Plans (SIPs) under the Clean Air Act, and so ensuring the accuracy of this emission inventory is important to the design of air quality policy for the SJV and California in general. In addition, previous studies have not taken advantage of the high-resolution
observations of $NH_{3(g)}$ made by the TES satellite instrument over Bakersfield during the CalNex campaign. Here we evaluate the consistency of the satellite, aircraft, and surface observations of $NH_{3(g)}$ and $NH_{4(p)}$ during the CalNex campaign and then use these observations, along with lidar retrievals of PBL height, to investigate the biases in the magnitude and diurnal cycle of emissions of $NH_{3(g)}$ from the CARB inventory in the SJV. We also explore the sensitivity of modeled $NH_3$ concentrations to bi-directional $NH_3$ exchange using the bi-directional $NH_3$ flux scheme in CMAQv5.0.2.
Section 2 briefly describes the data sources used in this study, while Section 3 describes the CARB emission inventory and the configurations used for the WRF, the Hybrid Single-Particle Lagrangian Integrated Trajectory (HYSPLIT), and CMAQ model runs. The performance of the CARB inventory used in our CMAQ simulations, along with model sensitivity studies, is presented in Section 4. Section 5 discusses the remaining errors in our final model configuration in detail and makes suggestions for further model improvements, while our conclusions are discussed in Section 6.

**2 Data**

**2.1 NOAA WP-3 aircraft**

The NOAA WP-3 aircraft completed 18 research flights during the CalNex campaign, which included measurements of $NH_{3(g)}$ and $NH_{4(p)}$. $NH_{3(g)}$ was measured at 1 s (~100 m) intervals using chemical ionization mass spectrometry (CIMS) with an uncertainty of +/- 30 % as described in detail in Nowak et al. (2007). The CIMS instrument sampled air through a 0.55 m
long heated teflon inlet with a fast flow. Measurement artifacts were accounted for by quantifying and subtracting the background signal originating from $NH_3$ desorption from instrument surfaces. The background signal was determined in flight by actuating a teflon valve at the inlet tip once every half hour to divert the sample air through a scrubber that removes $NH_3$ from the ambient air stream (Nowak et al, 2007). Additionally, standard addition calibrations from a $NH_3$ permeation tube were performed several times each flight to determine instrument sensitivity. Submicron $NH_{4(p)}$ was measured at 10 s (~
1 km) intervals with an uncertainty of ~ 30 % using a compact time-of-flight aerosol mass spectrometer from Aerodyne (c-TOF AMS, Bahreini et al., 2009). In this study we focused on the flights of 24 of May and 16 and 18 of June when the WP-3 was sampling air in the SJV (Figure 1). The quality-controlled flight data were reported at a merged time resolution of 1 s,

which we averaged to 1 minute values (the approximate time it takes the WP-3 to cross a 4 km CMAQ grid box) and then matched the sample times and locations to the corresponding time and location of the CMAQ hourly concentration output.

## 2.2 Bakersfield surface observations

Bakersfield, California is located in the southern part of the SJV (35.35°N, 118.97°W, 20 m asl) and there is a general north-to-south orographic air-flow in this region, with a tendency for emissions to get trapped in the valley due to the nearby mountains (Baker et al., 2013). At the Bakersfield ground site the Ambient Ion Monitor Ion Chromatograph (AIM-IC, Ellis et al., 2010, Markovic et al., 2012) was used to measure $NH_{3(g)}$ on an hourly basis, with an uncertainty of +/- 20 % and a detection limit of 41 ppt. The sampling inlet for the AIM-IC consists of an enclosure mounted at 4.5 m above ground, including a virtual impactor, parallel plate denuder, and particle supersaturation chamber, connected to the ion chromatography systems via several 20 m perfluoroalkyl sampling lines carrying the dissolved analytes (Markovic et al., 2014). This design reduces artifacts by minimizing the inlet surface area prior to scrubbing the $NH_3$ from the gas phase in the denuder, and by separating the gas and particle phase constituents while the sample flow is still at ambient temperature and relative humidity (Markovic et al., 2012). In addition, size-resolved, sub-micron non-refractory $NH_{4(p)}$ measurements were taken at 5 minute intervals using an Aerodyne Aerosol Mass Spectrometer (AMS, Liu et al., 2012). We averaged these data to 1 h time resolution in order to compare to the hourly CMAQ model output, which allowed for the evaluation of the ability of CMAQ to simulate the diurnal cycle of $NH_3$ concentrations. When $NH_{4(p)}$ measurements are available, we compare model results to $NH_x$ to reduce our sensitivity to gas-to-particle partitioning errors in the model; otherwise we compare to $NH_{3(g)}$.

## 2.3 TES $NH_3$ retrievals

During CalNex, TES made special observations (transects) near the Bakersfield, CA surface site with a horizontal separation of 12 km on six different afternoons. TES is a nadir-viewing Fourier-transform infrared (FTIR) spectrometer with a high spectral resolution of 0.06 cm$^{-1}$ and a nadir footprint of 5.3 km x 8.3 km. TES flies aboard the NASA Aura spacecraft, which is in a sun-synchronous orbit with an equator crossing time around 01:30 and 13:30 local solar time. Beer et al. (2008) reported the first satellite observations of boundary layer $NH_{3(g)}$ using the TES instrument. Shephard et al. (2011) developed and tested a full $NH_{3(g)}$ retrieval algorithm. The retrieval is based on an optimal estimation approach that minimizes the differences between the TES Level 1B spectra and a radiative transfer calculation that uses absorption coefficients calculated with the AER line-by-line radiative transfer model LBLRTM (Clough et al., 2006). The a priori profiles and covariance matrices for TES $NH_3$ retrievals are derived from GEOS-Chem model simulations of the 2005 global distribution of $NH_3$.

The TES $NH_{3(g)}$ retrievals generally have a region of maximum sensitivity between 700 hPa and the surface. While the retrieval is performed on 14 pressure levels, the number of degrees of freedom for signal (DOFS) is generally not greater than one. Therefore at any given single profile level the retrieved volume-mixing ratio (VMR) of $NH_3$ is highly influenced by the a priori profile. Rather than attempting to analyse data from individual retrieval levels, it is often desirable to express

the retrieved information in a representation where the influence of the a priori is reduced and the information available is collapsed to a single point. To address this issue, Shephard et al. (2011) developed a Representative Volume Mixing Ratio (RVMR) metric for $NH_{3(g)}$ based on similar techniques used previously for $CH_4$ (e.g., Payne et al., 2009; Wecht et al., 2012; Alvarado et al., 2015) and $CH_3OH$ (e.g., Beer et al., 2008). This RVMR represents a TES sensitivity weighted average value

where the influence of the a priori profile is reduced as much as possible; it generally ranges from 20 % to 60 % of the retrieved surface value for $NH_{3(g)}$. The minimum detection level for TES $NH_{3(g)}$ retrievals is an RVMR of approximately 0.4 ppbv, corresponding to a profile with a surface-mixing ratio of about 1-2 ppbv (Shephard et al., 2011).

Pinder et al. (2011) showed that the TES $NH_3$ retrievals were able to capture the spatial and seasonal variability of $NH_3$ over eastern North Carolina and that the retrievals compared well with in situ surface observations of $NH_3$, while Alvarado et al.

(2011) showed that TES $NH_3$ retrievals can also capture the higher concentrations of $NH_3$ in forest fires in Canada. Sun et al. (2015) demonstrated that under optimal conditions (i.e., good thermal contrast and $NH_3$ amounts significantly above the TES level of detectability), TES $NH_3$ agreed very well with in situ aircraft and surface measurements taken in the California Central Valley during the DISCOVER-AQ 2013 campaign.

There are at least three issues that have to be considered when using $NH_3$ satellite profiles to evaluate model predictions: (a)

the vertical resolution of the satellite profile is substantially coarser than that of the model profile; (b) the DOFS for $NH_3$ are generally less than 1.0; and (c) the retrieved satellite profile reflects the influence of the choice of a priori profile (Rodgers and Connor, 2003). Thus, in order to use these TES observations to evaluate CMAQ model predictions of the concentrations of $NH_{3(g)}$, we first interpolate the hourly CMAQ $NH_3$ profile predicted for 13:00 local solar time (expressed as the natural logarithm of the mixing ratio) to the TES pressure grid. We then apply the TES observation operator to the interpolated

CMAQ $NH_3$ profile to derive a model TES profile ($x_{TES}$). Finally, we apply the sensitivity weighting to calculate the model RVMR ($CMAQ_{RVMR}$). This value represents the RVMR that would have been retrieved if (a) TES had sampled a profile identical to the CMAQ-simulated profile and (b) the retrieval errors due to jointly retrieved parameters, other model parameters, and instrument noise were negligible. The observation operator equation is

$$x_{TES} = x_a + \mathbf{A}(x_{CMAQ} - x_a) \tag{1}$$

and the RVMR is calculated as

$$CMAQ_{RVMR} = \mathbf{W} * x_{TES} \tag{2}$$

where $x_a$ is a vector of the TES a priori $NH_3$ concentrations, $\mathbf{A}$ is the averaging kernel matrix, $x_{CMAQ}$ is a vector of the interpolated CMAQ $NH_3$ values, and $\mathbf{W}$ is a weighting vector (Rodgers and Connor, 2003; Shephard et al., 2011). $\mathbf{W}$ basically weights each level according to the sensitivity of the TES instrument at that level. It is calculated by summing the

most significant rows of the averaging kernel at each level (see the appendix in Shephard et al., 2011 for details).

**2.4 PBL heights**

Several studies have used lidar observations of aerosol profiles to determine the height of the planetary boundary layer (PBL) by identifying regions of large gradients in aerosol concentrations with height (e.g., Tucker et al., 2009; Lewis et al., 2013; Scarino et al., 2014; Hegarty et al., 2015). Scarino et al. (2014) and Tucker et al. (2009) define the mixed layer
measured by the HSRL as 'the volume of atmosphere in which aerosol chemical species emitted within the boundary layer are mixed and dispersed'. The NASA Langley Research Center (LaRC) airborne HSRL measured mixed layer heights during the CalNex campaign and the Carbonaceous Aerosol and Radiative Effects Study (Scarino et al., 2014), both of which we used in this study.

**3 Models**

**3.1 WRF-ARW**

CMAQ v5.0.2 was driven with meteorology provided by WRF ARW Version 3.5 (Skamarock and Klemp, 2008) that was configured with 3 nested domains of 36, 12, and 4 km horizontal grid spacing and 41 vertical layers. Shortwave and longwave radiation were calculated using the Rapid Radiative Transfer Model code for General Circulation Model applications (RRTMG, Mlawer et al., 1997; Iacono et al., 2008). The YonSie University (YSU, Hong et al., 2006) non-local
turbulent PBL scheme and the Noah land surface scheme (Chen and Dudhia, 2001) were used. Initial and boundary conditions for WRF were provided by the North American Regional Reanalysis (NARR, Mesinger et al*.,* 2006), which is recognized as state-of-the-science for North America (Bukovsky and Karoly, 2007). The WRF runs were 32-hour simulations initialized every 24 hours at 0000 UTC with analysis nudging of winds, temperature and humidity above the PBL on the outer 36 km domain only, as in Nehrkorn et al. (2013). The WRF outputs for UTC hours 09:00 to 32:00 from
each consecutive simulation were combined to form a continuous time series and the initial 8 hours of each simulation were discarded as spin-up time. The 8-h spin-up time and 32-h simulation length is longer than the 6-h spin-up time and 30-h simulation length used by Nehrkorn et al. (2013), but were necessary to perform 24-hour daily CMAQ runs using the 24-h daily CARB emissions files that started at 8:00 UTC. The WRF output was then converted to CMAQ-model-ready files using the Meteorology-Chemistry Interface Processor version 4.2 (MCIP).

**3.2 CMAQ**

We ran CMAQ on the inner 4 km WRF domain using the SAPRC07 chemical mechanism (Hutzell et al., 2012, Carter et al., 2010ab), which corresponds to the model-ready emission files for CalNex provided by CARB, and with the CMAQ AERO6 aerosol module with aqueous chemistry. Biogenic emissions, photolysis rates, and deposition velocities were all calculated inline. There were few clouds in California during this study period and thus lightning $NO_X$ emissions were negligible;
however, lightning $NO_x$ emissions were also calculated inline in CMAQ. Initial and horizontal boundary conditions for

CMAQ were provided by GEOS-Chem simulations on a 2° x 2.5° latitude-longitude grid for May and June 2010 following the approach of Lapina et al. (2014).

CMAQ emissions inputs for the state of California were provided as model-ready files by CARB, which prepared them using the Modeling Emissions Data System on a 4 km x 4 km grid-scale (available at http://orthus.arb.ca.gov/calnex/data/calnex2010.html, last accessed June, 2016). The emission change log is provided at ftp://orthus.arb.ca.gov/pub/outgoing/CalNex/2010/modelready/Change Log for Posted Inventories.pdf (last accessed June, 2016). In this inventory, the $NH_3$ emissions in SJV are assumed to be constant throughout the day (i.e., no diurnal cycle), and are constant day-to-day in a given month. While emissions do vary month-to-month, we do not explore seasonal variation in this study, since the measurement campaign only occurred during the months of May and June. As the CARB model-ready files had no out-of-state emission sources, our initial simulations were run using the CARB emissions for California, the GEOS-Chem boundary conditions, and no out-of-state emissions. We quantified the potential error in gas-phase $NH_{3(g)}$, Aitken and Accumulation mode aerosol $NH_{4(p)}$, and $NH_x$ in the SJV from neglecting out-of-state agricultural $NH_3$ emissions by using the agricultural $NH_3$ emissions from the NEI2011 platform, which we re-gridded from 12 km to our model's 4 km scale while keeping California state emissions constant. We performed this sensitivity test for a 7-day case study between 25-31 May with a 4-day spin up. Adding these out-of-state emissions had a negligible impact on the modeled $NH_3$ concentrations in the SJV (less than 0.001 % change), as the prevailing winds are mostly out of the north and northwest. Additionally, we tested the effect that errors in the boundary conditions from GEOS-Chem might have on the model runs. Doubling $NH_3$ boundary conditions for the same 7-day case study also had little impact on $NH_3$ concentrations in the SJV (less than 0.001 % change), which was expected based on the short lifetime of $NH_3$.

Finally, we also ran CMAQv5.0.2 using the bi-directional $NH_3$ flux scheme as developed by Bash et al. (2013) that uses fertilizer application data, crop type, soil type, and meteorology from MCIP output to calculate soil emissions potential and $NH_4$ to simultaneously calculate $NH_3$ deposition and emission fluxes for the CMAQ US domain. This scheme uses the U. S. Department of Agriculture's Environmental Policy and Integrated Climate (EPIC) model (Cooter et al., 2012) as contained in the Fertilizer Emissions Scenario Tool (FEST-C).

In order to evaluate CMAQ v5.0.2 modeled $NH_3$ in the SJV we ran three different scenarios for a month long case-study that covers the record of the Bakersfield surface observations (May 22 – June 22, 2010). The model scenarios include: 1) a baseline model run (CMAQ$_{base}$), in which the model was set up as described above, utilizing the CARB emissions inventory; 2) CMAQ$_B$, which ran with the baseline set up but also included the bi-directional $NH_3$ scheme described in Section 3.2, and finally 3) CMAQ$_{AB}$, which included both the bi-directional $NH_3$ scheme and diurnally-varying emissions in the SJV, as described in Section 4.1.

### 3.3 HYSPLIT

In order to explore the sources influencing the Bakersfield concentrations we ran the HYSPLIT model. Using meteorological inputs from the WRF 4 km domain discussed in Section 3.1, we generated 36-hour back trajectories with Version 4 of the

HYSPLIT model (Draxler and Hess, 1998) initiated from 100 m above ground level (agl) at Bakersfield at 17:00 PDT on June 18[th] back to 20:00 PDT on June 17[th]. Results from these runs are briefly discussed in Section 4.1 and shown in Figure S1.

## 4 Model Evaluation

The following subsections describe the evaluations of all three-model scenarios using the three different measurement datasets from the CalNex campaign. Section 4.1 describes the modeled evaluation using surface measurements, Section 4.2 using the aircraft measurements and finally Section 4.3, utilizing the TES satellite measurements.

### 4.1 Evaluation of modeled transport and diurnal variability of $NH_{3(g)}$ using surface observations

Table 1 shows that the $CMAQ_{base}$ scenario has a $NH_x$ positive mean bias (MB) of 8.24 ppbv and a mean normalized bias
(MNB) of 72.5% over the month-long surface data record; we focus on $NH_x$ so as to minimize the effects of possible model errors in gas-to-particle partitioning on our analysis, as discussed later in this section. $NH_3$ has a slightly higher bias, with $NH_{4(p)}$ having a lower MB of -0.40 ppbv, which has a small influence on total $NH_x$. However, this bias is not constant throughout the day, as can be seen in the $CMAQ_{base}$ results (blue line) shown in Figure 2. Figure 2a shows the average hourly ratio of $CMAQ_{base}$ modeled $NH_x$ versus measured concentrations for the Bakersfield ground site, averaged over all days of
the CalNex campaign; these ratios are derived from the boxplots shown in Figure 2b. The model bias shows a clear diurnal cycle, with $CMAQ_{base}$ significantly overestimating surface $NH_x$ concentrations at night by up to a factor of 4.5 and generally underestimating $NH_x$ during the daytime by a factor of 0.6 between 13:00 and 14:00 local time, consistent with the average $TES_{RVMR}$ observations near Bakersfield at about 13:30 local solar time, which is plotted as the green dot in Figure 2a and further discussed in Section 4.3. These results suggest that constant daily agricultural $NH_3$ emissions in the CARB inventory
(blue line Figure S2 in the Supplemental Material) may be misrepresenting the observed diurnal emission patterns. This is consistent with previous work done in North Carolina; Wu et al. (2008) found that $NH_3$ emissions from livestock feed lots show a strong diurnal cycle, peaking at midday.

Besides errors in emissions another contributing factor to the modeled bias of $NH_{3(g)}$ could be errors in gas-to-particle partitioning of $NH_{3(g)}$ to $NH_{4(p)}$. Figure 2a also shows that there is very little difference between the $NH_x$ (solid blue) and
$NH_{3(g)}$ (dashed-blue) lines, indicating only a small fraction of total $NH_{3(g)}$ is converted into $NH_{4(p)}$ in this region, consistent with Baker et al. (2013). Thus, errors in gas-particle partitioning of $NH_3$ in CMAQ, while important for accurately estimating $PM_{2.5}$ concentrations, cannot account for the diurnal errors in $NH_x$ we have observed.

Another potential source of diurnal errors in modeled $NH_x$ are diurnal variations in meteorology, which could alter the source regions to which the Bakersfield site was sensitive throughout the day. Differences between modeled and true $NH_3$
emission errors at upwind sites would thus appear as diurnal errors in $NH_x$. We ran a HYSPLIT case study for June 18[th], where back trajectories were run for eight different times during the day (Figure S1). During the CalNex campaign, the

daytime flow is generally from the north/north-west that is funnelled through the California Central Valley towards Bakersfield. During the nighttime there is a shift in wind direction to sources coming from the southeast. Cooling air from up in the eastern mountain ranges causes a mountain drainage effect into the southern valley area. This interaction of the mountain drainage combined with the typical low-level jet from the northern central valley creates a *Fresno Eddy*, as described in Michelson and Bao (2008). Figure 3 shows a wind rose for all points included in Figure 2, where measured wind direction and $NH_x$ concentrations are shown on the left, and modeled wind direction and $NH_x$ concentrations are shown on the right. It can be seen in Figure S5a that the nighttime wind measurements from the southeast generally have lower wind speeds ($< 4$ m s$^{-1}$) and that the model does not capture the variation of these wind speeds very well. This may be due to some timing errors in that the model may not capture true winds within a 4 km grid box, which corresponds to about 1-2 hours in real time. In general, many of the higher modeled $NH_x$ concentrations appear to be occurring during nighttime when the model should have winds out of the southeast, thus there is large model bias for these points. As indicated by the performed HYSPLIT back-trajectories, and the description of air flow in the southern valley, we assume that although the measurements indicate the immediate wind direction was out of the south-east, the air-mass's long-range transport still travelled over the Central Valley to accumulate emissions from that region before being recirculated by the Fresno Eddy to eventually come from the southeast. Thus, an overestimate of emissions in the Central Valley at night could still contribute to a model overestimate of measurements coming out of the southeast, rather than this air mass having come from a *cleaner* source, east of the mountains. Additionally, for the remaining time periods and majority of measurements not out of the southeast at nighttime, the model does a better job at simulating wind speeds (Figure S3), with a large model bias in $NH_x$ concentrations remaining. Thus diurnal changes in transport are likely not the only contributing factor to the diurnal mismatch shown in modeling results.

Diurnal errors in the PBL height estimates could also be responsible for the diurnal error pattern in the CMAQ $NH_x$ concentrations at Bakersfield (Figure 2). We used daytime HSRL measurements taken in the SJV during CalNex to evaluate our WRF simulated PBL heights. Figure 5 shows 2-minute averages of the HSRL calculated mixed layer height compared to the WRF PBL for three daytime flights that passed over the SJV. The modeled and measured heights show good agreement, with a slope of 0.76, $r^2$ of 0.70, and mean bias of 87 m. Thus errors in daytime PBL height do not seem to account for much of the underestimate in modeled daytime $NH_x$. Scarino et al. (2014), when comparing all CalNex HSRL flight measurements to their configuration of the WRF-Chem model, found similar results. In summary, gas-to-particle partitioning and PBL height errors are likely not responsible for the diurnally varying measurement to model biases.

CARB $NH_3$ emissions in the SJV are constant both diurnally and day-to-day, with an hourly flux of around 0.23 moles s$^{-1}$ for the Bakersfield area (Figure S2). The Bakersfield ground measurements, however, indicate there should be a diurnal pattern of lower emissions at night and higher emissions during the day, as has been previously reported of $NH_3$ emissions from livestock (e.g., Bash et al., 2013; Zhu et al., 2015a) and other agricultural $NH_3$ sectors (Skjøth et al., 2011). The intense agricultural activities in the SJV generate large $NH_3$ emissions, with concentrations often exceeding 5 ppb as indicated in the ground measurements, making this an *NH₃ rich* region relative to the ambient sulfate concentrations. In this regime, since

there is not enough sulfate to react with all the $NH_3$, a simple box model over the Bakersfield site, with wind speed, deposition, and PBL height variation held constant, would show a linear relationship between additional $NH_3$ emissions and the $NH_x$ concentration (Seinfeld and Pandis, 2006). Thus we expect errors in other parameters (PBL height, deposition, etc.) to affect modeled $NH_{3(g)}$ and $NH_x$ concentrations to a greater degree, and we investigate these parameters below.

To test our hypothesis that the diurnal errors in $NH_x$ concentrations are due to diurnal errors in $NH_3$ emissions we explored two additional model scenarios to attempt to improve the diurnal cycle of $NH_3$ emissions in the CMAQ model. We found that including the bi-directional flux of $NH_x$ in the $CMAQ_B$ case (green lines) significantly reduces the nighttime concentration peaks of ground-site measured $NH_3$. However there is still a clear model $NH_x$ overestimate overall (MB of 4.57 ppb and large MNB of 45.74 %, see Table 1), and the low correlation is not improved ($r^2 = 0.01$). The $CMAQ_B$ scenario

also shows overestimates following the day's maximum in temperature (Figure 4). At night this bias is reduced relative to the total concentrations.

We then applied a scaling factor to all $NH_3$ area sources per grid box in the SJV, based on the $CMAQ_{base}$ bias relative to the ground measurements. To do this, we first calculated the total $NH_3$ area source emissions for each grid box, based on additional information on the emissions breakdown from the CARB inventory. For Kern County, where Bakersfield, CA

resides, pesticide/fertilizer applications dominate the $NH_3$ emissions inventory at 72%, followed by farming operations (that include handling of all livestock and excrement) at 25%, and other sources for the remaining fraction. Table S2 in the Supplemental Material describes the fraction of $NH_3$ emissions for counties in the SJV. We then calculated the emissions for each hour based on the hourly average ground measurements and considering the *NH_3 rich* conditions. Note that the adjusted maximum emissions vary by about a factor of 4.5 from the minimum at night to the midday peak, as can be seen in Figure

S1 (solid red line) which is more modest than the factor of 10 variation seen in livestock feedlots (Bash et al., 2013; J. Bash, personal communication, Oct. 6, 2015). We then reran CMAQ with both these adjusted emissions and the bidirectional $NH_3$ scheme (the $CMAQ_{AB}$ run) to assess the impact. Despite applying the scaling factor to all emissions instead of solely to the feedlots as in Bash et al. (2013), the $CMAQ_{AB}$ model predictions, shown as the purple lines in Figure 4, matches the measurements (black line) better than the $CMAQ_{base}$ or $CMAQ_B$ scenarios over the day and night, with large outliers

seemingly reduced, consistent with Bash et al. (2013). The mean nighttime bias for $CMAQ_{AB}$ was reduced by about a factor of 2 and the overall bias of $NH_x$ reduced to -1.23 ppbv (Table 1); this model version does particularly well between the hours of 01:00 and 06:00 (see Figure 2a). The fact that adding the diurnally-varying emission profile reduces the model bias, even though the emissions are dominated by fertilizer applications that should be accounted for by the bi-directional $NH_3$ scheme, suggests that the bi-directional $NH_3$ scheme in CMAQ v5.0.2 is not correctly accounting for the diurnal variations in $NH_3$

flux in the SJV. Furthermore, when we compare the modeled $NH_3$ to measured values coming from just the southeast at night (Figure S5), the model bias is reduced by about factor of 3.5. This suggests that although the model may not capture the immediate wind direction and wind speed at night, as explained above, because of the long-range transport down the Central Valley that evolves into the Fresno Eddy, reducing emissions in this upwind region also reduces model bias for these

points in time. However, we note that the correlation of all three-model scenarios remains very low ($r^2 < 0.06$), suggesting further model errors, such as the neglect of any day-to-day variation in $NH_3$ emissions in our simulations.

As noted above, the results for $NH_{3(g)}$ generally track the results for $NH_x$ already discussed. In contrast, the model usually under-predicts the small amount of $NH_{4(p)}$ observed (on average < 1 ppbv, Figure 4c) by a factor of 2, with little variation

between the model scenarios (Table 1, MB of $NH_x$ for $CMAQ_{base}$, $CMAQ_B$ and $CMAQ_{AB}$ of -0.40, -0.41 and -0.44 respectively). These model errors in $NH_{4(p)}$ reflect not only model errors in total $NH_x$, but also errors in the formation of $HNO_{3(g)}$ and $SO_{4(p)}$ (Figure S3). $HNO_{3(g)}$ is overestimated in all model simulations up to a factor of 4, with concentrations not changing between model cases. $SO_{4(p)}$ measured concentrations are minimal and do not appear to have any trend and also do not change with model cases. However, as our interest in this study is in constraining $NH_3$ emissions, not inorganic aerosol

formation, we do not investigate these errors further here.

## 4.2 Evaluation of modeled vertical distribution of $NH_{3(g)}$ using aircraft observations

The aircraft observations in the SJV indicate a large underestimate (range of factors about 1 to 5) in $CMAQ_{base}$ modeled $NH_x$ concentrations above the surface, as shown in Table 2 (all flights in SJV) and Figure 6 (two flights). The variation in model concentrations in the background of Figure 6 are due to the aircraft flying in and out of different horizontal grid boxes in the

model. The May 24[th] flight shows a strong $CMAQ_{base}$ $NH_x$ underestimate of about a factor of 5 when considering the entire flight with a low correlation ($r^2$) of 0.31 and a mean bias of -1.95 ppbv. This significant underestimate could potentially be due to an underestimate of vertical mixing at night (discussed below); when only data before 18:00 PDT is considered (assuming this is before the collapse of the convective boundary layer) the underestimate is only a factor of ~1.5 and the $r^2$ is 0.77, a considerably better and statistically significant result. However, model comparisons to flight data on 16 and 18 of

June before 18:00 PDT, likely before the boundary layer collapse on these days, show a significant model underestimate and low $r^2$ values, thus there may be other contributing factors to this bias and lack of correlation, such as errors in vertical transport and the neglect of day-to-day variability in the emissions.

A daytime versus evening flight measurement evaluation of $CMAQ_{base}$ shows a clear difference in the vertical distribution of $NH_x$. At night (May 24[th] flight, Figure 6b), the model contains most of the $NH_x$ in the lowest model level, whereas during the

day (June 16[th] flight) it vertically mixes the $NH_x$ (Figure 6a). Based on the higher $NH_x$ concentrations that the aircraft is measuring these results could suggest 1) vertical mixing is stronger than simulated in the model during both day and night flights or 2) that there is a residual layer of $NH_x$ at night that is not captured by the model or 3) there is a non-local source that is also not well captured by the model.

Gas-phase $NH_3$ can either be deposited to or emitted from the surface depending on the land-type, land-use, and ambient

concentrations (Bash et al., 2015; Fowler et al., 2009). The $CMAQ_{base}$ run does not take this into consideration, but when bi-directional $NH_3$ is calculated with a diurnal emission factor included in $CMAQ_{AB}$, $NH_3$ dry deposition should generally decrease, increasing the net land-atmosphere flux (Bash et al., 2013). The $CMAQ_{AB}$ model run shown in Figure 6c is consistent with these results (and inconsistent with the hypothesis that vertical mixing is underestimated in the model) as the

vertically distributed concentration of $NH_x$ significantly increases from the $CMAQ_{base}$ case to the $CMAQ_{AB}$ case. The transport of $NH_3$ also seems to increase, this being a potential explanation for the plume entering the plot domain around 21:00 PDT in the bottom curtain plot. The total column concentration of $NH_x$ also increases, leading to a significant positive model bias for the $CMAQ_{AB}$ scenario (e.g. in the earlier part of the flight in Figure 6c and Table 2), suggesting a possible

overestimation of total $NH_x$ emissions by the bi-directional $NH_3$ scheme and further enhanced by adding a diurnal emission factor during the afternoon and evening hours when the flights took place. This indicates that the diurnal factor application in $NH_3$ emissions at the surface grids does not significantly change the concentrations aloft, where the flight measurements are taking place compared to the $CMAQ_{AB}$ case, resulting in remaining model bias and requiring further investigation.

### 4.3 Evaluation of modeled $NH_{3(g)}$ with TES $NH_3$ retrievals

Applying the TES operator to the CMAQ profiles and calculating the $CMAQ_{RVMR}$ allows us to compare the satellite and model datasets quantitatively, as described in Section 2.3. Surface $NH_3$ from the $CMAQ_{base}$ run (Figure 7a) and the TES $NH_3$ RVMR (Figure 7b) along a sample TES transect both identify the regions of large $NH_3$ sources and the spatial changes along the transect and demonstrate that the $CMAQ_{RVMR}$ is underestimated for the base run, particularly at higher $NH_3$ RVMRs. Similar results were found for other transects and summarized in Table 3 and Table S2. The time of the satellite overpass

occurs just prior to the peak of emissions in the emission factor applied to the $CMAQ_{AB}$ case which in turn increases the RVMR bias to 1.31 ppbv and increases the regression slope to 1.02 (purple line Figure 8) as compared to a bias of -2.57 and slope of 0.40 in the $CMAQ_{base}$ case. The slope of the linear regression of $CMAQ_{AB}$ RVMR suggests that CMAQ run with bi-directional ammonia along with the applied emissions factor slightly overestimates $NH_3$ concentrations, indicating the magnitude of the emissions factor may be too high at the time of satellite over pass. The inclusion of the emission factor in

this $CMAQ_{AB}$ case has a higher bias than the bi-directional model run, $CMAQ_B$. This demonstrates the importance of using highly time-resolved observations of $NH_3$ to determine the diurnal cycle of $NH_3$ along with polar-orbiting satellite retrievals of $NH_3$ to improve the spatial and seasonal distribution of the emissions, as noted in Zhu et al. (2013). In other words, if we had relied solely on the TES observations at 13:30 local solar time to evaluate the $CMAQ_{base}$ runs, we would have incorrectly assumed that the CARB inventory was a factor of 2.4 too low for total $NH_3$ emissions, whereas the surface data demonstrate

that the problem is primarily in the diurnal cycle of the emissions.

Modeled RVMR can be very sensitive to errors in the modeled vertical distribution of $NH_3$. We investigated this by comparing each level of the TES retrieved $NH_3$ profile with the corresponding CMAQ profile level after the observation operator is applied. Figure 9 shows box-and-whisker plots of this comparison for the $CMAQ_{base}$ and $CMAQ_{AB}$ model scenarios ($CMAQ_B$ not shown). This plot differs from that in Shephard et al. (2015) in that it includes the average of layers

below 908 mb, which introduce an RVMR bias due to levels that are below 1000 mb. The $CMAQ_{AB}$ case shows the smallest bias of the three modeled scenarios in the lowest pressure level (~1 ppb) with the higher levels showing little bias as well (~0.08 ppb). Thus comparing the TES and CMAQ profiles level-by-level indicates that the $CMAQ_{AB}$ scenario demonstrates

the least bias in simulating the TES retrievals, consistent with the conclusions based on the surface observations in Section 4.1.

## 5 Discussion

The results in Section 4 show that the $CMAQ_{AB}$ model scenario that included both the bi-directional $NH_3$ scheme and the
diurnally adjusted emissions provided results that were much closer to the surface measurements (Section 4.1) and satellite (Section 4.3) observations than the $CMAQ_{base}$ runs, with measurement uncertainties explained in Section 2. The $CMAQ_{AB}$ simulations did result in a large overestimate of $NH_x$ concentrations higher in the atmosphere as measured by the aircraft (Section 4.2). Here we discuss the remaining errors in the $CMAQ_{AB}$ scenario, suggest possible explanations for these errors, and make suggestions for the direction of future research.
Model bias in both the night and daytime simulation of surface $NH_x$ is reduced in the $CMAQ_{AB}$ scenario. The total bias is significantly reduced from the factor 4.5 at night and 0.6 during the day compared to the $CMAQ_{base}$ scenario (Figure 4a). In $CMAQ_{AB}$, the model does well between the hours of 1:00 am and 6:00 am local time (Figure 2a), perhaps related to the lower emissions at this time of day when adjusted emissions are used assuming the linear relationship of emissions to concentrations. The remaining diurnal bias shows a relative model underestimate with a factor of ~0.6 at 10:00 local time
and a relative model overestimate peaking at ~1.7 at 19:00 local time (Figure 4c), with average $CMAQ_{AB}$ modeled concentrations slightly higher in the afternoon and peaking around 19:00 (Figure 4d). It is interesting to note that the $CMAQ_{AB}$ bias relative to surface concentrations is small near the TES overpass time (e.g., crossing 0% between 13:00 and 14:00 local time, Figure 4c), which is consistent with the small bias seen in the comparison with the TES observations in Section 4.3. Furthermore, the aircraft results for the $CMAQ_{AB}$ scenario discussed in Section 4.2 also show a large relative
overestimate in the afternoon and evening when the flights took place (Table 2), consistent with the afternoon and evening overestimates seen in the surface data.

Thus all three datasets suggest that the remaining errors in modeled $NH_x$ concentrations may be due to the diurnal profile of the net land-atmosphere $NH_3$ flux in the $CMAQ_{AB}$ run peaking too late in the day. One possibility to this is that the diurnal cycle we applied to the non-fertilizer $NH_3$ emissions, which was based on the ambient measurements of $NH_3$, is peaking too
late in the day. However, as the peak of our assumed diurnal profile for these emissions (Figure S1) is consistent with the peak in surface temperature (1:00 pm, Figure 4d), we consider this explanation less likely than remaining errors in the bi-directional $NH_3$ scheme for fertilizer emissions.

These errors in the bi-directional $NH_3$ scheme could be due to errors in the dynamic emissions response of the bi-directional $NH_3$ scheme to local temperature, wind direction and speed (Bash et al., 2013). However, Figure 4d shows that the modeled
surface temperature and wind speed are not that far off from the values observed at the Bakersfield site for the majority of measurements out of the northwest, and for those out of the southeast that are not captured in the model, we believe that the long-range transport of these winds through the Central Valley prior to entering the Fresno Eddy are dominating the

emissions profile of that air mass, thus influencing the final concentration of that air mass. Thus the remaining errors are less likely related to errors in atmospheric meteorological conditions, and are more likely due to errors in the land-air interactions and the dependence of soil conditions (e.g., soil temperature, pH, and water content) on meteorology and crop management practices as calculated within the bi-directional $NH_3$ scheme (Cooter et al., 2012). The scheme calculation assumes two soil

layers (0.01 m and 0.05 m) that independently exchange $NH_x$ with the canopy, which then exchanges $NH_x$ with the surface layer of the atmosphere (Bash et al., 2013). If the calculation of the response of soil properties in these layers to surface meteorology and crop management practices is incorrect (e.g., the soil layers do not heat up or cool down quickly enough with the change in surface temperature), that would affect the amount of $NH_x$ available from the soil as well as the rate at which the soil $NH_4^+$ is converted to $NO_3^-$ through nitrification (Bash et al., 2013). This would result in errors in the flux of

$NH_x$ from the soil to the canopy, thus altering the canopy compensation point and the net atmospheric flux.

The aircraft results may also suggest errors in the vertical mixing of $NH_x$ during the afternoon and evening (e.g., the peak of the PBL height and the collapse). While we consider this effect as likely less important to the remaining errors in $CMAQ_{AB}$ than the potential errors in the bi-directional $NH_3$ scheme already discussed, an overestimate of vertical mixing during the afternoon would overestimate the flux of $NH_x$ from the surface layer of the atmosphere to the upper levels, reducing the

concentrations, which is consistent with the aircraft overestimate. In addition, the soil-canopy-surface atmosphere system would respond to this overestimate of vertical mixing by increasing the net flux of $NH_x$ from the soil to the atmosphere in order to maintain equilibrium, resulting in a total overestimate of the emissions of $NH_x$ during the afternoon and evening.

We thus recommend that future work to improve the simulation of atmospheric $NH_x$ concentrations in the SJV focus on bottom-up and top-down approaches that will better estimate the diurnal changes in the canopy compensation point that

determines the net flux from the land to the atmosphere in the bi-directional $NH_3$ scheme (Bash et al., 2013). This scheme was originally developed using field scale observations taken in North Carolina, USA (Walker et al., 2013), so it is not surprising that this approach may require modifications to work in the SJV. We recommend, first, that the CARB $NH_3$ inventory be updated to better separate $NH_3$ emissions from fertilizer and livestock sectors. The Bash et al. (2013) scheme assumes that these two sectors will dominate $NH_3$ emissions, while the CARB inventory divides fertilizer/pesticide use from

"farming operations", thus it is unclear if these other farming practices are dominated by livestock or not. Second, crop management data (e.g., fertilizer amount, timing, form, and distribution) used in EPIC (and thus in the CMAQ bi-directional $NH_3$ scheme) are based on data for the entire West Coast of the US (e.g., California, Oregon, and Washington), and thus may not be representative of farming practices in the SJV. Better crop management data specific to the SJV, as well as more SJV-specific data on soil moisture and heating rates, may thus help in removing some of the remaining errors in the $CMAQ_{AB}$

scenario. Third, in order to better connect these bottom-up emission estimates to the measured atmospheric concentrations, we recommend that top-down studies focus not just on correcting the net $NH_x$ flux to the atmosphere but also determine the diurnally-varying biases in the canopy compensation point that determines these net fluxes. This may require the development of adjoint methods and models (e.g., Zhu et al., 2015a) that can retrieve time-varying correction factors for the canopy compensation point, rather than just for the net flux itself.

**6 Conclusions**

We used $NH_3$ retrievals from the NASA Tropospheric Emission Spectrometer, as well as surface and aircraft observations of $NH_{3(g)}$ and submicron $NH_{4(p)}$ gathered during the CalNex campaign, to evaluate the ability of the CMAQ model run with the CARB emission inventory to simulate ambient $NH_{3(g)}$ and $NH_{4(p)}$ concentrations in California's San Joaquin Valley. We find

that CMAQ simulations of $NH_3$ driven with the CARB inventory are qualitatively and spatially consistent with TES satellite observations, with a correlation coefficient ($r^2$) of 0.64. However, the surface observations at Bakersfield indicate a diurnally varying model bias and low correlation, with $CMAQ_{base}$ overestimating $NH_3$ at night by at times more than 50 ppbv and underestimating it during the day by up to 10 ppbv. The surface, satellite, and aircraft observations all suggest that the afternoon $NH_3$ emissions in the CARB inventory used in $CMAQ_{base}$ are underestimated by at least a factor of two, while the

nighttime overestimate of $NH_3$ is likely due to a combination of overestimated nighttime $NH_3$ emissions and underestimated nighttime deposition. Thus the diurnally-constant $NH_3$ emissions used by CARB in the SJV appear to misrepresent the diurnal emission cycle.

Using the bi-directional $NH_3$ scheme in CMAQ ($CMAQ_B$) resulted in reduced $NH_x$ concentrations at night and a slight increase during the day, overall reducing the model bias relative to the surface and satellite observations. However, this

scenario substantially increased the simulated mixing ratio of $NH_x$ at higher altitudes, leading to an increased bias relative to the aircraft observations. In addition, errors in the simulation of the nighttime surface concentrations remained in this scenario.

In order to evaluate the diurnal impact of $NH_3$ emissions, we used the surface observations at Bakersfield to derive an empirical diurnal cycle of $NH_3$ emissions in the *NH_3 rich* region of the SJV in which nighttime and midday emissions

differed by about a factor of 4.5. Despite the model not capturing winds out of the southeast at night, adding a diurnal profile to the CMAQ bi-directional $NH_3$ simulations ($CMAQ_{AB}$) while keeping the daily total $NH_3$ emissions constant at the CARB values significantly reduced the model bias at night relative to the surface observations, on top of the already reduced bias from the $CMAQ_B$ simulations. Comparisons with the TES RVMR showed a slight increase in the bias for the $CMAQ_{AB}$ scenario relative to $CMAQ_B$, but further examination of the modeled and retrieved vertical profiles suggests that this is

primarily due to ~1 ppb differences in the lowest retrieved level with the $CMAQ_{AB}$ scenario showing little bias (0.08 ppbv) relative to the TES $NH_3$ profile above this surface level. However, despite nighttime reduction in model bias in the $CMAQ_{AB}$, scenario sizable errors (up to 20 ppbv) in the afternoon and evening $NH_3$ and low model correlations remained, possibly due to the net land-atmosphere $NH_3$ flux calculated by the bi-directional $NH_3$ scheme peaking too late in the day due to errors in the calculated response of the soil conditions (e.g., soil temperature, pH, and water content) to meteorology

and crop management practices.

We recommend that future work on modeling $NH_x$ emissions in the SJV include (a) updating the CARB $NH_3$ inventory to account for $NH_3$ from fertilizer, livestock, and other farming practices separately, (b) adding information on crop management practices specific to the SJV region to the EPIC-FESTC system, and (c) top-down studies that focus not just on

correcting the net $NH_x$ flux to the atmosphere but also on determining the diurnally-varying biases in the canopy compensation point that determines these net fluxes.

## Acknowledgements

The authors thank Leo Ramirez and Jeremy Avise at CARB for the emissions inventory data they provided, as well as Eli Mlawer, Thomas Nehrkorn and Elizabeth Steinhubel of AER and Jesse Bash, Jim Kelly, and Kirk Baker of the EPA for their valuable comments and discussions on this work. This work was funded by the NOAA Climate Program Office Atmospheric

Chemistry, Carbon Cycle, & Climate (AC4) program through Grants NA130AR4310060 and NA140AR4310129 to CRL, JDH, KCP, and MJA of AER and DKH, MDT, and SLC of CU Boulder. Development and evaluation of the TES $NH_3$ retrieval was funded through NASA grants to KCP of AER such as Grant NNH08CD52C. LMR acknowledges support from California Air Resources Board (CARB) for funding the AMS measurements (contract 09–328). Infrastructure support for the Bakersfield ground site was provided through CARB contract 08-316 to Ron C Cohen and Allen H Goldstein at the

University of California, Berkeley. The collection and analysis of CalNex surface observations of $NH_{3(g)}$ at Bakersfield were funded by a University of Toronto Centre for Global Change Science award to MM and a NSERC graduate scholarship to TCV of the University of Toronto. We also wish to thank the NOAA P3 aircraft flight crew and technicians. The conclusions of this paper are the authors' only, and do not reflect NOAA or CARB policy.

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

Table 1. Summary statistics of the modeled $NH_x$, $NH_{3(g)}$ and $NH_{4(p)}$ concentration comparisons to the ground measurements for all three model runs. Mean Bias (MB) = mean (modeled – measured), Mean Normalized Bias (MNB) = mean ([modeled – measured]/measured). Note that low $r^2$ values, less than 0.10, are highlighted in italics.

| Model Run | NHx Slope | $r^2$ | MB (ppbv) | MNB (%) | NH_{3(g)} MB (ppbv) | MNB (%) | NH_{4(p)} MB (ppbv) | MNB (%) |
|---|---|---|---|---|---|---|---|---|
| CMAQ_base | -2.49+/-0.15 | *0.001* | 8.24 | 72.54 | 8.63 | 78.79 | -0.40 | -52.96 |
| CMAQ_B | 1.22+/-0.07 | *0.01* | 4.57 | 45.74 | 4.99 | 50.60 | -0.41 | -55.92 |
| CMAQ_AB | 0.85+/-0.05 | 0.05 | -1.23 | -10.70 | -0.79 | -14.01 | -0.44 | -60.24 |

5  Table 2. Summary statistics of the modeled to measured $NH_x$ concentration comparisons following the SJV flights. Mean Bias (MB) = mean (modeled – measured), Mean Normalized Bias (MNB) = mean ([modeled – measured]/measured). Note that low $r^2$ values, less than 0.10, are highlighted in italics.

| Date | Time (PDT) | NHx Slope | $r^2$ | MB (ppbv) | MNB (%) | NH_{3(g)} MB (ppbv) | MNB (%) | NH_{4(p)} MB (ppbv) | MNB (%) |
|---|---|---|---|---|---|---|---|---|---|
| CMAQ_base | | | | | | | | | |
| | 16:00-22:00 | 0.20+/-0.01 | 0.31 | -1.95 | -2.010 | -1.74 | -18.24 | -0.14 | -58.70 |
| 20100524 | 16:00-18:00 | 0.68+/-0.05 | 0.77 | -0.20 | -10.79 | -0.04 | -32.46 | -0.08 | -53.19 |
| | 18:00-22:00 | 0.18+/-0.01 | 0.29 | -2.40 | -0.213 | -2.24 | -14.65 | -0.14 | -60.10 |
| 20100616 | 13:00-18:00 | 0.30+/-0.02 | 0.43 | -5.92 | -8.980 | -4.90 | -3.59 | -0.24 | -45.32 |
| 20100618 | *13:00-18:00* | *0.18+/-0.02* | *0.10* | *-8.12* | *-18.97* | *-7.85* | *-28.9* | *-0.26* | *-75.20* |
| CMAQ_B | | | | | | | | | |
| | 16:00-22:00 | 0.36+/-0.03 | 0.09 | 5.56 | 351.82 | 5.71 | 453.86 | -0.10 | -39.32 |
| 20100524 | *16:00-18:00* | *-1.57+/-0.24* | *0.19* | *6.59* | *506.18* | *6.71* | *639.07* | *-0.07* | *-31.92* |
| | *18:00-22:00* | *0.31+/-0.03* | *0.11* | *5.30* | *31.28* | *5.46* | *407.1* | *-0.11* | *-41.18* |
| 20100616 | 13:00-18:00 | 0.76+/-0.06 | 0.04 | 6.27 | 248.03 | 6.63 | 279.85 | -0.22 | -33.82 |
| 20100618 | 13:00-18:00 | 0.37+-0.04 | 0.02 | 4.26 | 394.88 | 4.41 | 458.88 | -0.21 | -52.37 |
| CMAQ_AB | | | | | | | | | |
| | 16:00-22:00 | 0.38+/-0.03 | 0.17 | 6.15 | 369.73 | 6.30 | 474.89 | -0.10 | -38.48 |
| 20100524 | 16:00-18:00 | -1.61+/-0.25 | 0.16 | 6.94 | 526.88 | 7.07 | 664.26 | -0.07 | -31.17 |
| | 18:00-22:00 | 0.32+/-0.02 | 0.22 | 5.95 | 330.05 | 6.10 | 427.07 | -0.11 | -40.33 |
| 20100616 | *13:00-18:00* | *0.80+/-0.06* | *0.10* | *7.83* | *264.1* | *8.19* | *297.58* | *-0.22* | *-33.83* |
| 20100618 | *13:00-18:00* | *0.42+/-0.05* | *0.03* | *5.59* | *425.7* | *5.76* | *494.16* | *-0.21* | *-50.36* |

Table 3. Summary statistics of the $CMAQ_{RVMR}$ to $TES_{RVMR}$ $NH_3$ comparisons for 4 CalNex overpasses (05/28, 05/30, 06/13, 06/15). Mean Bias (MB) = mean (modeled – measured), Mean Normalized Bias (MNB) = mean ([modeled – measured]/measured).

| Model Run | Slope | $r^2$ | MB (ppbv) | MNB (%) |
|---|---|---|---|---|
| $CMAQ_{base}$ | 0.47 | 0.64 | -2.57 | -30.21 |
| $CMAQ_B$ | 0.93 | 0.60 | 0.84 | 14.40 |
| $CMAQ_{AB}$ | 1.02 | 0.60 | 1.31 | 19.57 |

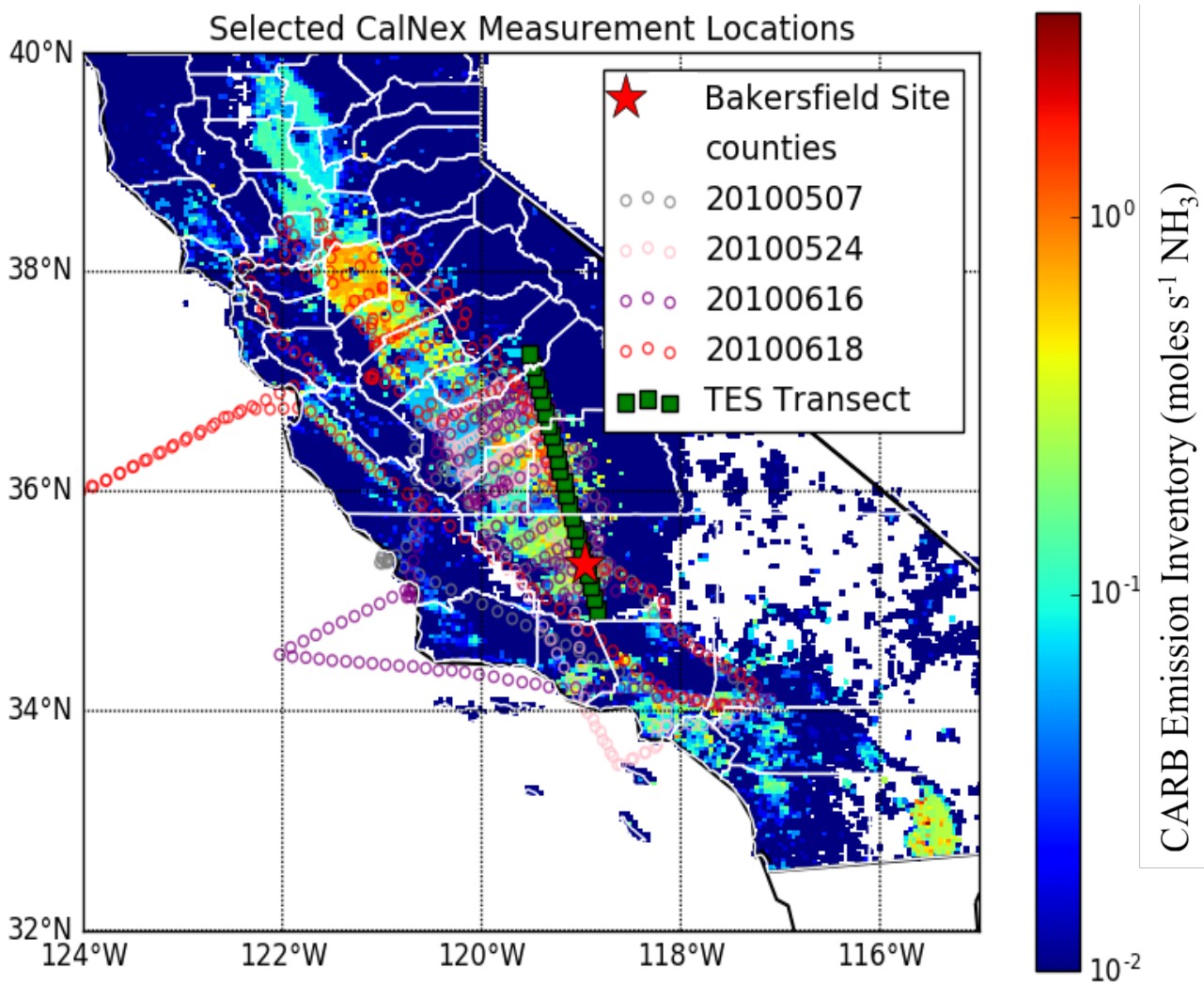

Figure 1. Distribution of NH$_3$ emissions across California (background) on May 12, 2010 at 19:00 UTC as well as P3 flight tracks (small circles), TES transect (green squares), and the Bakersfield site (red star) with the county lines shown in white.

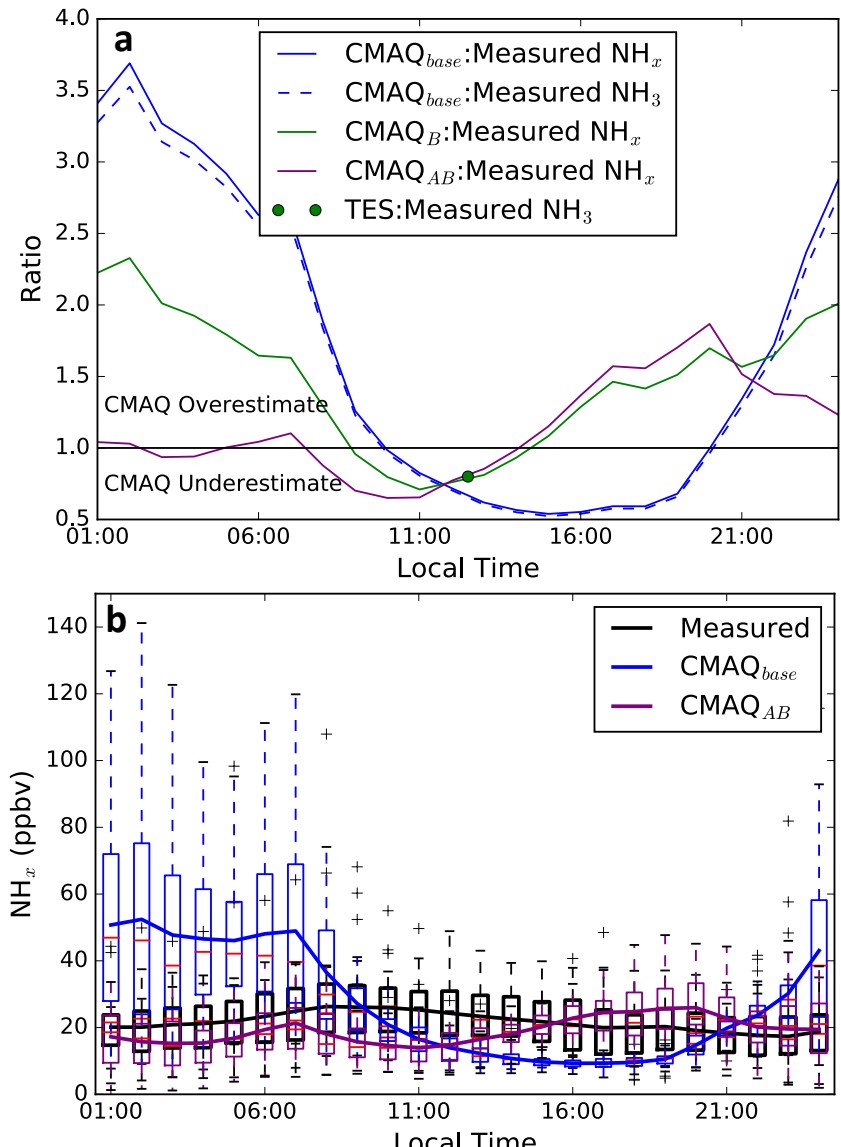

Figure 2. (a) The average hourly ratio of modeled to measured $NH_3$ (dashed line) and $NH_x$ (solid line) mixing ratios at the Bakersfield ground site for the $CMAQ_{base}$(blue), $CMAQ_B$(green) and $CMAQ_{AB}$ (purple) cases, and the average modeled RVMR to TES RVMR ratio (green dot) in local time. (b) Boxplot of average hourly $NH_x$ mixing ratios at the Bakersfield ground site for the measured (black), $CMAQ_{base}$ (blue) and $CMAQ_{AB}$ (purple) cases, averaged over all measurement days during CalNex where the boxplots show the inter-quartile range and median line (red) within the box and outliers (whiskers), with the solid lines showing the mean for that day.

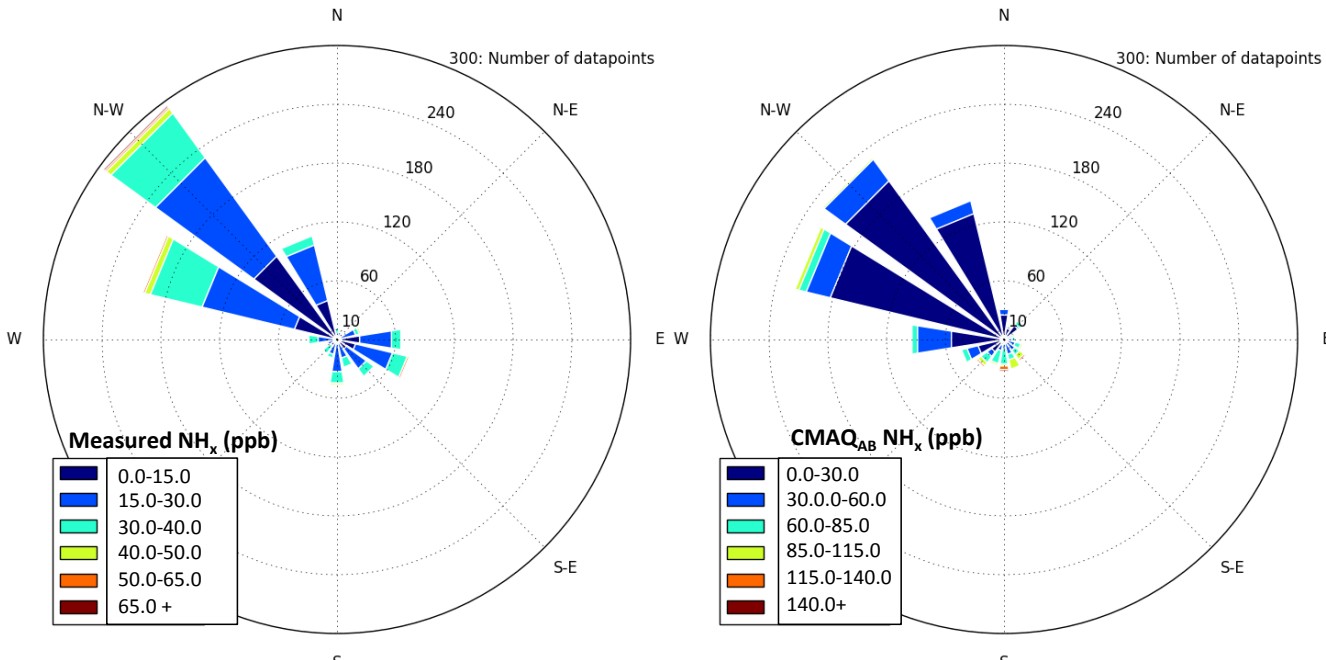

Figure 3. Wind rose of measured wind direction and $NH_3$ on the left, and $CMAQ_{AB}$ modeled wind direction and $NH_3$ on the right where contours represent number of data points (hourly) per wind direction. Note the difference in scale, where values are in ppb.

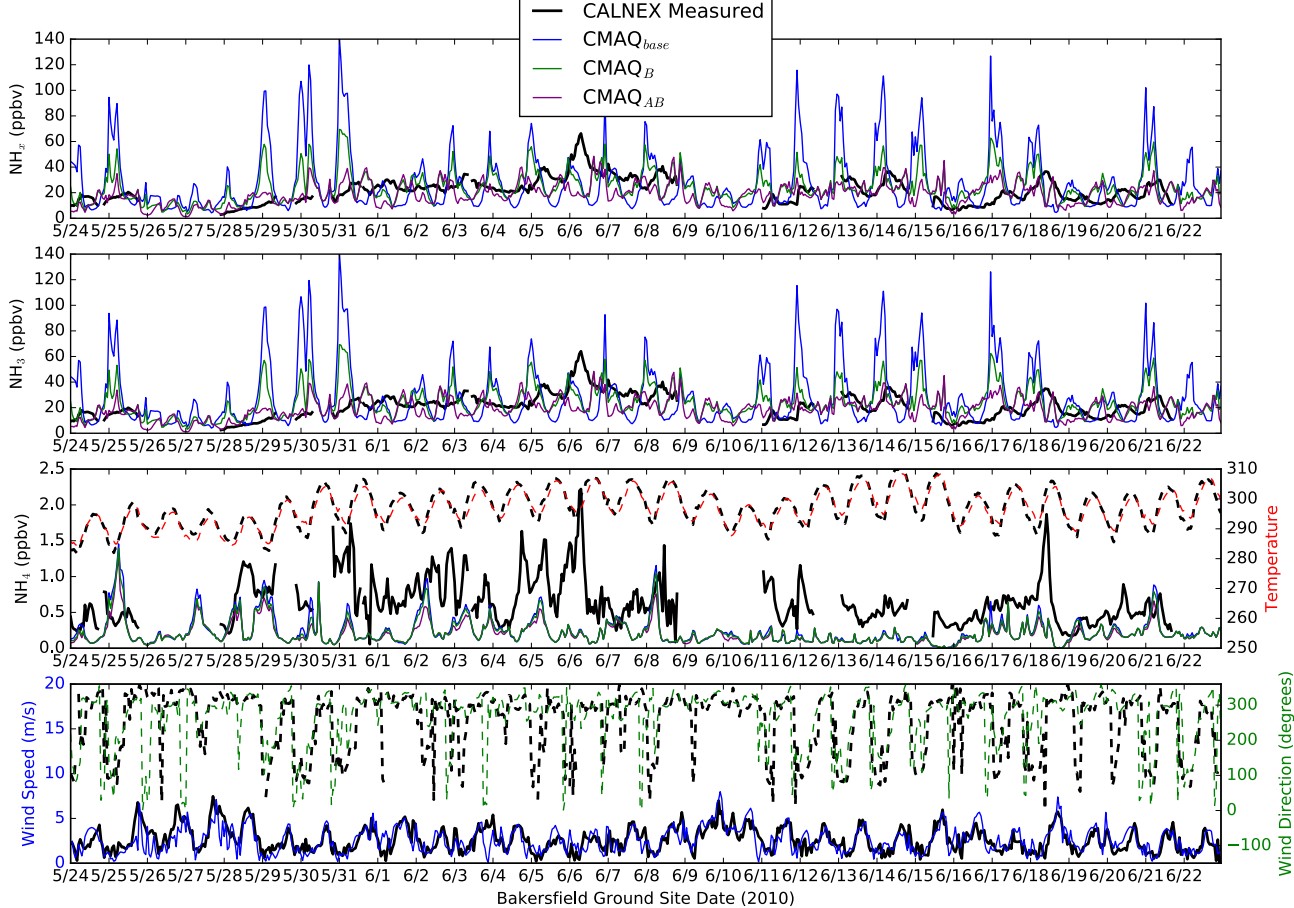

Figure 4. The CalNex ground measurements at the Bakersfield site (solid black) compared to the CMAQ<sub>base</sub> (solid blue), CMAQ<sub>AB</sub> (purple) and CMAQ<sub>B</sub> (green) simulations for a month of model runs. The top panel (a) shows NH<sub>x</sub>, b) shows NH<sub>3(g)</sub>, c) NH<sub>4(p)</sub>, and temperature (K) and d) wind speed on the left and wind direction on the right axis.

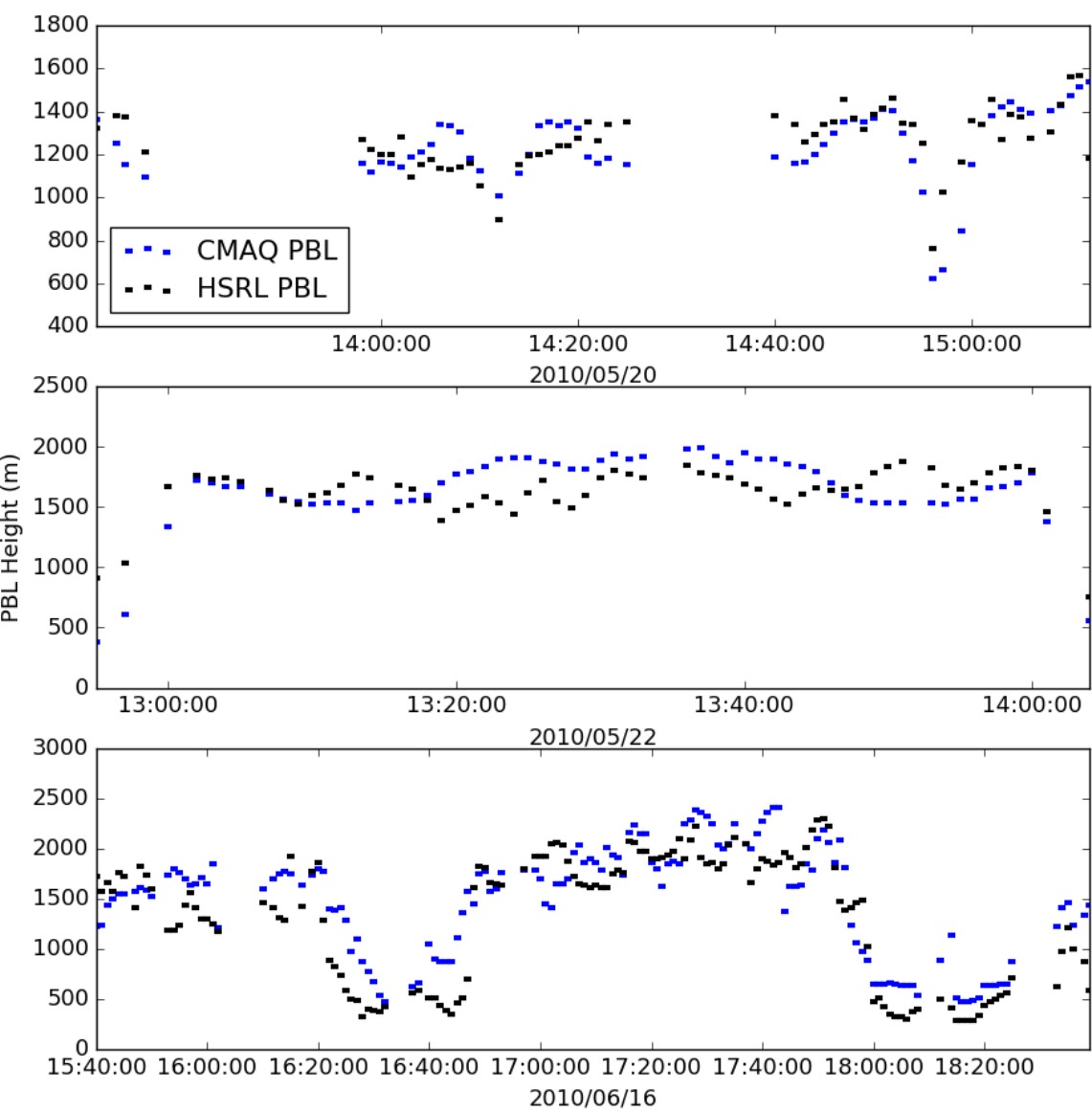

Figure 5. Time series of WRF predicted planetary boundary layer heights and HSRL calculated mixed layer heights for 3 flight sections in the San Joaquin Valley (2 during CalNex and one during a CARES campaign).

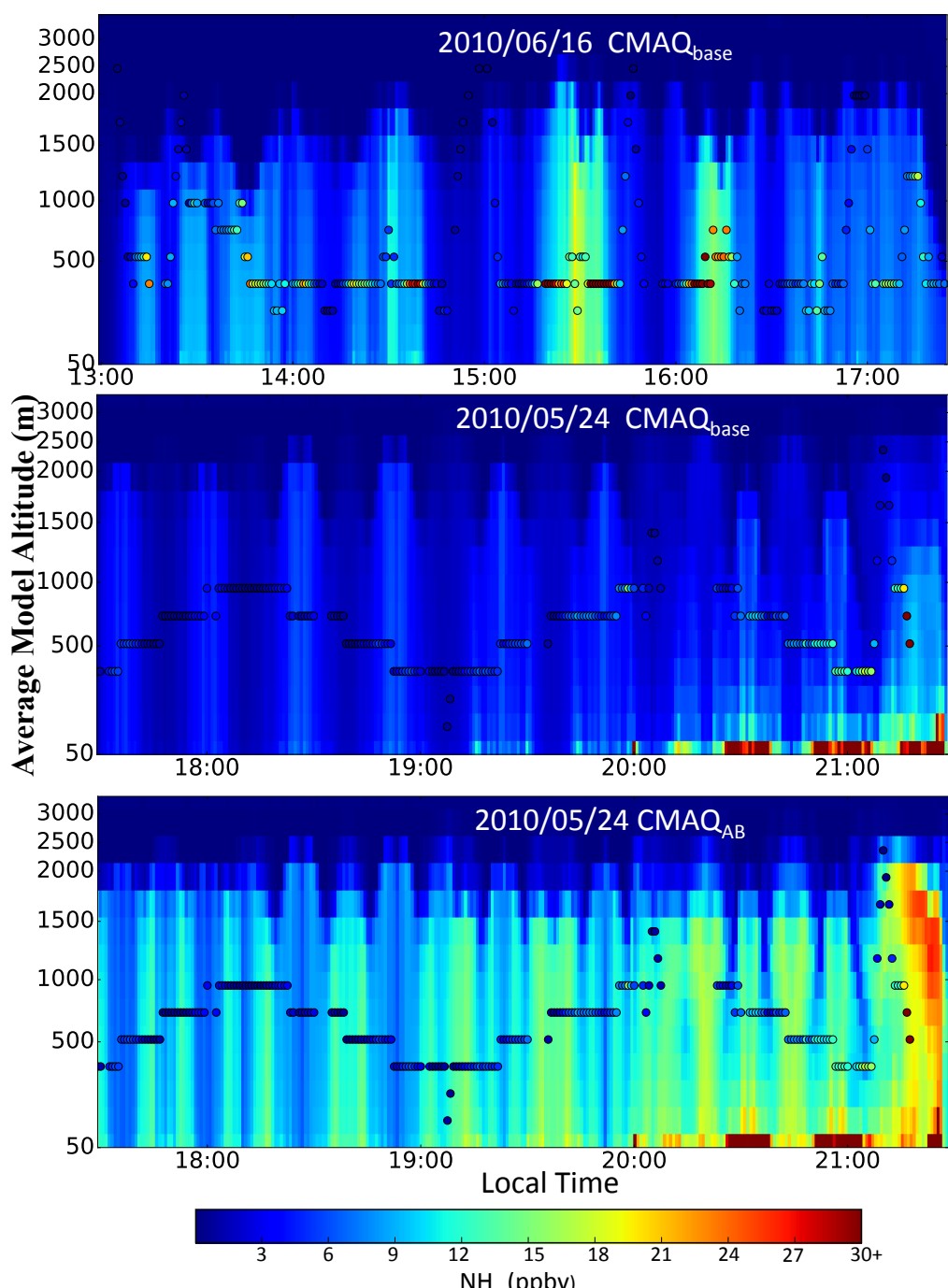

Figure 6. (a) The hourly output of $CMAQ_{base}$ $NH_x$ is shown in the background with the measured (one minute average) $NH_x$ concentrations within the modeled hour shown as the dots for the daytime flight on June 16, 2010 and (b) a nighttime flight on May 24, 2010, and (c) the same nighttime flight but for the $CMAQ_{AB}$ scenario.

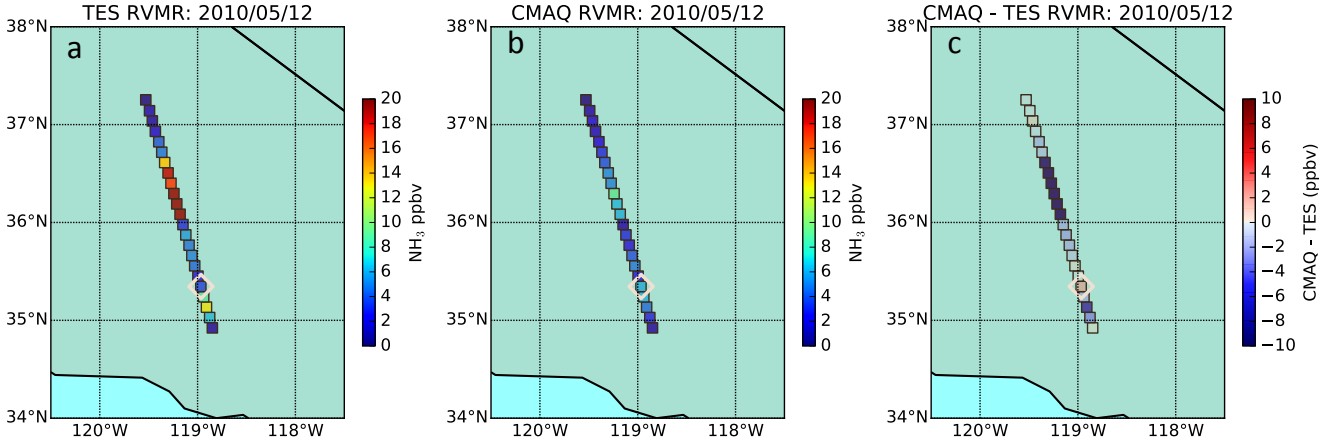

Figure 7. NH$_3$ representative volume mixing ratios (RVMRs) on 12 May 2010 during the CALNEX campaign for (a) TES special observations, (b) modeled RVMR for CMAQ and (c) the difference between each RVMR near the Bakersfield, CA, surface site with the white diamond locating the Bakersfield measurement site.

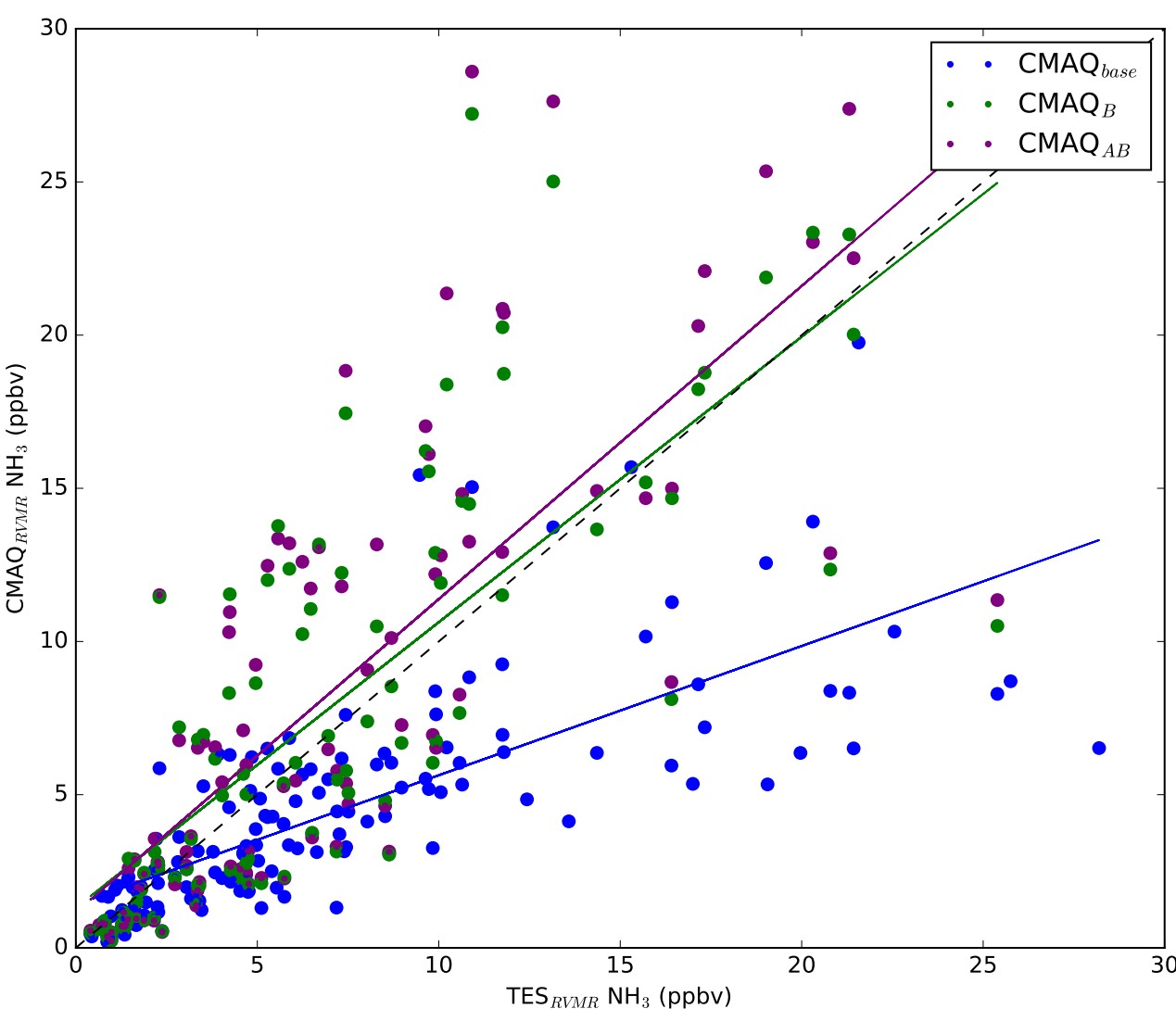

Figure 8. Scatter plot of CMAQ_base (blue), CMAQ_B (green) and CMAQ_AB (purple) versus TES NH₃ representative volume mixing ratios for TES special observation passes (TES_RVMR) during the CalNex campaign with statistics discussed in Table 3.

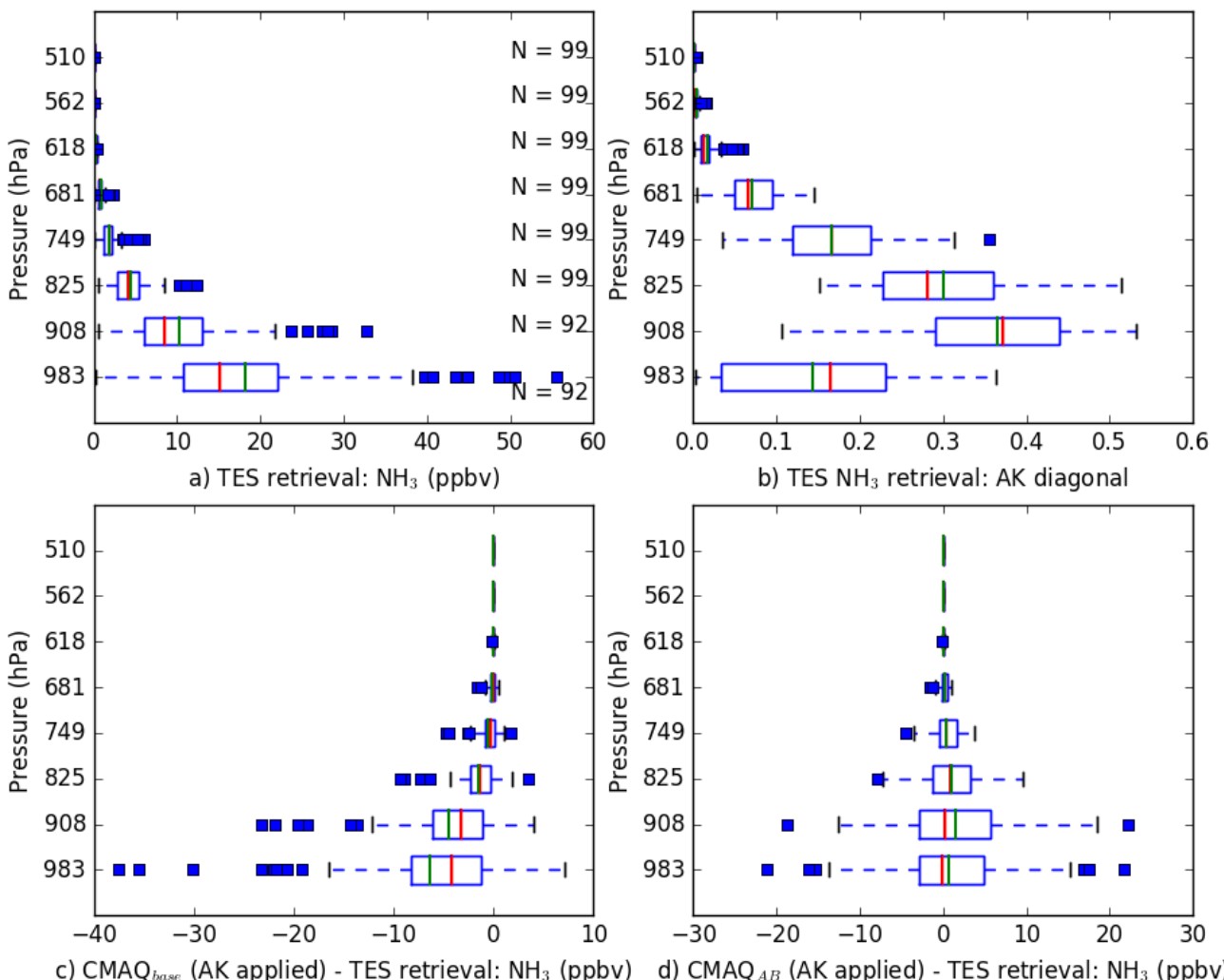

Figure 9. Boxplots of a) TES NH$_3$ retrieval by pressure level, b) TES NH$_3$ retrieval averaging kernel (AK) diagonal, c) difference between the TES NH$_3$ retrieval and CMAQ$_{base}$ modeled NH$_3$ interpolated to TES levels with an AK applied for the baseline model run and d) same as Panel c but for the CMAQ$_{AB}$ run. Box plots show the mean (green), median (red), interquartile range (IQR, blue box), whiskers at 1.5 IQR and outliers beyond that.