# Peer review of "Modeling the Diurnal Variability of Agricultural Ammonia in Bakersfield, California during the CalNex Campaign"

_Atmospheric Chemistry and Physics, 2016_

## Referee Comment (RC1) · Anonymous Referee #1 · 4 Apr 2016

General comments: The authors present a NH3 modelling study based on a variety of ammonia measurements (surface, flight, satellite) performed during the CalNex measurement campaign. The focus lies on determining possible causes for the discrepancies in the diurnal variation between the modelled and observed concentrations. The paper further describes a few possible adjustments by applying a more representative temporal allocation of the NH3 emissions and the application of a bi-directional NH3 flux scheme.

An interesting study is presented, comparing modelled concentrations with NH3 observations covering a range of the vertical and spatial distribution of ammonia concentrations. Potentially this could be a really useful and informative publication, as almost no model studies using this range of observations are available, but some work will be needed to improve it before final publication.

[Figure]

In particular, I found that with the wealth of data and observations available more could be done, specifically on the presentation of the results and a systematic (final) discussion which is more or less missing. The setup of the paper is done rather well, describing the observations used as well as the applied models etc. The results & discussion/conclusion section however will need some work. As an example the last section feels a bit rushed. In the first paragraph of the results the comparison is described in a systematic fashion, while for the final version of the model only a short description is given, lacking any final conclusions, which leaves the reader without any sense of improvement/idea that the final version of the model improved much but the bias(a bit).

Specific comments:

I am missing an overall figure with the observed and modelled concentrations for the Bakersfield concentrations. The authors do show the diurnal cycles and a boxplot for the individual hours but this does not give a feel of the possible events and variability between the days which can occur during the measurement interval. A simple plot with the time series would bring some clarity. One could also add the observed TES observations as a second Y-axis. Another idea would be to add temperature/wind speeds to explain the variation of the concentrations(as emissions from fields for example are related to both).

2. On the measurements itself: Surface: If the instruments used have an inlet with some piping etc, this could cause artifacts in the observed $NH_3$ concentrations in the early morning. Some words on this and other possible artifacts would be helpful.

Only seven days of the observations are used and compared with the model, is this the entire measurement period? If not why are only 7 days of the measurements used?

Satellite: I am not convinced by the model vs satellite comparison. Especially the comparison for the observations near Bakersfield look rather poor. Some words on the quality of the TES data? Also by using the RVMR on has to know for certain the vertical

is described well in the model. The RVMR is only 20%-60% of the surface value, and depending on the NH3 profile doesn't have to be in relation to the surface. Some words on any effects caused by the RVMR and maybe a comparison on the profile of TES and the model? Similar to the study by shephard et al?: Shephard, M. W., McLinden, C. A., Cady-Pereira, K. E., Luo, M., Moussa, S. G., Leithead, A., Liggio, J., Staebler, R. M., Akingunola, A., Makar, P., Lehr, P., Zhang, J., Henze, D. K., Millet, D. B., Bash, J. O., Zhu, L., Wells, K. C., Capps, S. L., Chaliyakunnel, S., Gordon, M., Hayden, K., Brook, J. R., Wolde, M., and Li, S.-M.: Tropospheric Emission Spectrometer (TES) satellite observations of ammonia, methanol, formic acid, and carbon monoxide over the Canadian oil sands: validation and model evaluation, Atmos. Meas. Tech., 8, 5189-5211, doi:10.5194/amt-8-5189-2015, 2015.

Also the observed concentrations near Bakersfield seem to be quite low at time. Any effects due to the sensitivity/retrieval of TES for these low retrieved concentrations? You could add a figure with the observed and modelled profiles and the AVK of the satellite observations to show the difference in the vertical (and yes the DOF are low but the profiles are still used for the RVMR). Aircraft observations: Possible artifacts? Include the uncertainties in the discussion of the results / conclusions.

2. Model: Some discussion on performance of the model for the vertical distribution of NH3 would be helpful. Also include some words on the performance for species like HNO3 and sulfates as these are probably causes for any discrepancies in the diurnal cycle.

2.3. PBL: Figure 7. shows the performance of the WRF PBL when compared to the HSRL observations. The authors conclude that the deviations are not a probably cause for any faults in the diurnal cycle of NH3. I do not agree with this conclusion. From the plot one can conclude that for small PBL heights there are large deviations up to a factor 2 when compared to the modelled WRF PBL. You can convince me by showing a figure of the diurnal cycle of the PBL for both HSRL and WRF? And/or the normalized version of the cycle? By adding a diurnal cycle figure you also strengthen

any comments and conclusions in the paper that the errors in the PBL have no effect.

2.x Emissions: Can you add a table or a short paragraph on the emissions sources and their relative totals?

3.3 Hysplit, I think this section can be removed as in the remainder of the paper only 3 sentences are dedicated to the results.

4. Results: A bit of extra structure and discussion in the results will greatly improve the manuscript. The authors have a wealth of data available but only scarcely use it. The flight data is only used for the basic version of the model, and not discussed in the latter parts of the manuscript, while the variation in the emissions will also affect the vertical distribution of the NH3 concentrations. The systematic discussion of possible causes for the discrepancies between the modelled and measured concentrations as given in 4.2 should be added for the other versions of the model. Each version should rule out one or more of the possible causes, which will add to the overall discussion of the state of the model (and not just this model, but the overall performance of most CTMs). Adding a table with the airborne observations vs each of the modelled versions would help.

Table 2. add some correlations and statistics similar to table 1. A figure or table in which you split the statistics per hour of the day will give some further insight on the performance of the model for each part of the day. Partially this is done already in figure 4, but some correlations / bias plot could be added for more information. In this figure/table one can then easily point out the improvements in the later model versions similar to figure 8.

5. Conclusion / Discussion:

I am missing a final discussion on how one would improve the model in the future or what kind of measurements would be needed (does not have to be long). A few points for a start of the overall discussion and state of CMAQ/NH3 modelling (bit broad):

- What kind of measurements would the authors perform to further understand the model and reasons for discrepancies between modelled and observed concentrations
- Discuss point by point what this study improved, for now I can only see a small improvement to the bias. - -

Final words:

I recommend rewriting some parts of the manuscript following a few of the stated highlights to improve the overall quality of the paper. When rewritten this paper can be a great start for future model (improvement) studies.

---

## Referee Comment (RC2) · Anonymous Referee #2 · 5 Apr 2016

General comments: This study presents a study combining surface, aircraft and satellite measurements of NH3 and NH4 concentrations in the San Joaquin Valley with a model study using CMAQ. The approach taken enables the authors to identify lacks in knowledge in both model description and emission inventories. While this is a worthwhile effort, the analysis and discussion could be improved by more explicitly including a discussion section in which the possible explanations of mismatch between model and observation are listed, as well as an outlook section with possible improvements to the model or emission data. If these points are improved upon (I give a few suggestions below), this paper could really contribute to improving NH3 modelling and to a better understanding of the sources of mismatch between model and measurements.

Specific comments: For readers not familiar with the SJV geography and the location of the Bakersfield site, providing a map of the region could be valuable.

[Figure]

Title: Not everyone is familiar with CalNex. Adding 'campaign' (or otherwise clarifying the term) at the end of the title would make it clearer.

Introduction: While the introduction presents a thorough overview of previous work, it is rather elaborate. Condensing this by focusing on the most important points would increase readability. This is later also true for the description of the TES data and the CMAQ model. Please specify at some point in which period the CalNex campaign was active.

Data: Please give coordinates and elevation of the Bakersfield site.

Models: In the text on the CMAQ model results of sensitivity studies are already provided. Consider moving this to the results section. You could also consider dedicating a paragraph to the description of the emission database (which are the most important ammonia sources, etc.) as this is so important in your uncertainty analysis later. Page 8, line 32: ' . . . soil emissions potential and NH4': sentence is incomplete.

Results: Section 4.1: You claim that the relative changes in NH3 concentration along the transect are captured well by the model, but to me this seems not to be the case: the highest concentrations are underestimated much more strongly, also in a relative sense, than the lower concentrations outside the direct source region. Also, based on figure 3 you conclude that CMAQ with the CARB inventory captures the spatial variability near Bakersfield well, but given the correlation coefficient of 0.22 for the overpasses closest to Bakersfield I'm not sure this statement holds. The purpose of highlighting these points is not clear as it is later not at all discussed. If I understand the plot and caption correctly, each point represents one overpass in one grid cell, i.e., this plot shows both temporal and geographical variability. Is this correct? If yes, could you comment on which part of the scatter is caused by temporal and which by geographical variability? Section 4.2: From figure 4 I don't see an underestimation of a factor 2.5 during the daytime, rather 1.5-2. Line 26-27 (page 9) would be better supported by adding a time series of the measurement-model comparison to show the seasonal patterns. Lines 3-6 (Page 10) seem redundant. Lines 13 and onwards (page 10): please mention that you now compare concentrations at 400+ meters above the surface; otherwise the step from the ground-based observations to air craft might be confusing. Page 10, line 28 – page 11, line 7: This section can be shortened significantly; consider if results that are not worth showing are worth talking about. Section 4.3: What does 'consistent with measured temperature patterns' mean? I assume it suggests that temperature is the driving variable for the emission variability during the day, could you state that more clearly? Why did you only adjust the hourly emission profile for NH3, was there no day-to-day variability (e.g. related to temperature) to take into account? With the approach taken, you assume that concentrations in a certain hour are dominated by emissions in that same hour; could you comment on this assumption? Why did you decide to test the new diurnal profile for 7 days only? A comparison to the aircraft data would be valuable here as well, to see to what extent the changed diurnal profile impacts modelled concentrations and model performance at higher altitudes.

Section 5: This section would be stronger if it contained more than a summary of the most important points of the paper, but also a discussion on future steps / important work to be done to improve the modelling of ammonia and the representation of emissions. For example, a discussion on the relative importance of the misrepresentation of emission diurnal cycles vs. misrepresentation of the vertical mixing (which should we work on first?) would be valuable.

Technical comments: Page 2, line 6: photoxidize should be photo-oxidize Page 3, line 11: CONUS might not be a known acronym for non-US readers; please explain. Page 3, line 25: Write out TES as it is the first mention in the main body of the article. In general: check for unexplained abbreviations. Page 7, lines 11 and 12: HSRL is mentioned as acronym but only written fully at the second instance. Page 8, line 14: SoCAB is not explained. Page 9, line 13: scatterplot should be scatter plot Page 11, line 23: remove 'mostly' as the CARB NH3 emissions are completely constant.

---

## Author Comment (AC1) · 24 Jun 2016

**Addressing Reviewer comments for "Modeling the Diurnal Variability of Agricultural Ammonia in Bakersfield, California during the CalNex Campaign"**

We thank both reviewers for their helpful comments and suggestions. We provide specific responses to their comments below that coincide with changes, additions and corrections made in the paper. We feel this has resulted in an improved manuscript. Below reviewer comments are in italics while our responses are in normal typeface.

**Reviewer #1**

*1) General comments: The authors present a NH3 modelling study based on a variety of ammonia measurements (surface, flight, satellite) performed during the CalNex measurement campaign. The focus lies on determining possible causes for the discrepancies in the diurnal variation between the modelled and observed concentrations. The paper further describes a few possible adjustments by applying a more representative temporal allocation of the NH3 emissions and the application of a bi-directional NH3 flux scheme. An interesting study is presented, comparing modelled concentrations with NH3 observations covering a range of the vertical and spatial distribution of ammonia concentrations. Potentially this could be a really useful and informative publication, as almost no model studies using this range of observations are available, but some work will be needed to improve it before final publication*
*In particular, I found that with the wealth of data and observations available more could be done, specifically on the presentation of the results and a systematic (final) discussion, which is more or less missing. The setup of the paper is done rather well, describing the observations used as well as the applied models etc. The results & discussion/conclusion section however will need some work. As an example the last section feels a bit rushed. In the first paragraph of the results the comparison is described in a systematic fashion, while for the final version of the model only a short description is given, lacking any final conclusions, which leaves the reader without any sense of improvement/idea that the final version of the model improved much but the bias (a bit).*

We thank the reviewer and agree with their comment. In order to address this concern we have rearranged the model results section so as to organize the paper around the three measurement platforms (ground, flight and satellite measurements) rather than around the different model sensitivity studies and added a new Discussion Section. Section 4.1 in the new manuscript evaluates a full month of output from the three different model scenarios, $CMAQ_{base}$, $CMAQ_B$ (bidirectional ammonia) and $CMAQ_{AB}$ (bidirectional and adjusted emissions), with the surface measurements. We thoroughly discuss the evaluation of the diurnal distributions of $NH_x$, $NH_{3(g)}$ and $NH_{4(p)}$ in the three scenarios, since the surface measurements provide the best available information for identifying diurnal patterns. Section 4.2 discusses the aircraft measurement comparisons as well as the vertical profile of $NH_x$, since the aircraft measurements provide insight into the vertical dispersion of $NH_x$ as well as the overall magnitude of emissions in the afternoon and evening hours Section 4.3 discusses the comparison of the three scenarios with the TES retrievals, including an added level-by-level comparison of the modeled and retrieved vertical profiles of $NH_3$. Section 5 then describes the remaining errors in the final model version ($CMAQ_{AB}$) and suggests possible explanations for these errors and directions for future research. The revised Conclusions Section (Section 6) summarizes the results and discussion, and now includes a description of the overall model bias changes as well as suggestions for future work on $NH_3$ modeling in the SJV.

*2) I am missing an overall figure with the observed and modelled concentrations for the Bakersfield concentrations. The authors do show the diurnal cycles and a boxplot for the individual hours but this does not give a feel of the possible events and variability between the days, which can occur during the measurement interval. A simple plot with the time series would bring some clarity. One could also add the observed TES observations as a second Y-axis. Another idea would be to add temperature/wind speeds to explain the variation of the concentrations (as emissions from fields for example are related to both).*

We have added a detailed time series plot (Figure 2 in the revised manuscript) that compares a full month of model output from our three model scenarios to ground measurements for $NH_x$, $NH_{3(g)}$, $NH_{4(p)}$, temperature, and wind speed. As the TES measurements are not directly comparable with the surface data we have not plotted them in Figure 2, but we feel that the additional figure (Figure 8 in the revised manuscript) and table (Table 3 of the revised manuscript) that we have added give additional context for the TES observations (see the response in comment 5 below). We also include a final model ($CMAQ_{AB}$) to measurement comparison in Figure 9 and discuss these results in the new Discussion Section (Section 5).

*3) On the measurements itself:*
*Surface: If the instruments used have an inlet with some piping etc, this could cause artifacts in the observed NH3 concentrations in the early morning. Some words on this and other possible artifacts would be helpful.*

We added Page 6, lines 12-17 (quoted below) to discuss how the measurement techniques address artifacts from inlets and other surface measurement uncertainties due to instrumentation and include an additional reference (Markovic et al., 2014).

> "At the Bakersfield ground site the Ambient Ion Monitor Ion Chromatograph (AIM-IC, Ellis et al., 2010, Markovic et al., 2012) was used to measure $NH_{3(g)}$ on an hourly basis, with an uncertainty of +/- 20 % and a detection limit of 41 ppt. The sampling inlet for the AIM-IC consists of an enclosure mounted at 4.5 m above ground, including a virtual impactor, parallel plate denuder, and particle supersaturation chamber, connected to the ion chromatography systems via several 20 m perfluoroalkyl sampling lines carrying the dissolved analytes (Markovic et al., 2014). This design reduces artifacts by minimizing the inlet surface area prior to scrubbing the $NH_3$ from the gas phase in the denuder, and by separating the gas and particle phase constituents while the sample flow is still at ambient temperature and relative humidity (Markovic et al., 2012)."

*4) Only seven days of the observations are used and compared with the model, is this the entire measurement period? If not why are only 7 days of the measurements used?*

Our original paper only covered a 7-day case study and we felt that a longer time period would provide a better evaluation. We have thus significantly expanded our three model scenarios to cover the entire month-long period of ground measurements at the Bakersfield supersite and added extensive discussion of these results to the paper. Page 9, lines 30-31 and Page 10, lines 1-5, in Section 4 describe the full range of model runs as follows:

> "In order to evaluate CMAQ v5.0.2 modelled $NH_3$ in the SJV we ran three different scenarios for a month long case-study that covers the record of the Bakersfield surface observations (May 22 – June 22, 2010). The model scenarios include: 1) a baseline model run ($CMAQ_{base}$), in which the model was set up as described in Section 3.2, utilizing the CARB emissions inventory; 2) $CMAQ_B$, which ran with the baseline set up but also included the bi-directional $NH_3$ scheme described in Section 3.2, and finally 3) $CMAQ_{AB}$, which included both the bi-directional $NH_3$ scheme and diurnally-varying emissions in the SJV, as described in Section 4.1. The following subsections describe the evaluations of all three model scenarios using the three different measurement datasets (surface, aircraft, and satellite) from the CalNex campaign."

We decided against re-running our sensitivity studies ($CMAQ_B$ and $CMAQ_{AB}$) to start at the initial model start of May 5[th] as we felt that a full month of comparison to ground measurements was sufficient to evaluate the two additional model scenarios ($CMAQ_B$ and $CMAQ_{AB}$) while minimizing the additional computational expense this would have caused. This time frame also included the majority of flight days and TES overpasses, and the average bias relative to TES for

$CMAQ_{base}$ did not change significantly when excluding the two overpasses prior to May 22. We still present the comparisons of the $CMAQ_{base}$ model run with the two flights and TES overpasses prior to May 22[nd] in the Supplemental Material (Table S1 and S3).

*5) Satellite: I am not convinced by the model vs satellite comparison. Especially the comparison for the observations near Bakersfield look rather poor. Some words on the quality of the TES data? Also by using the RVMR on has to know for certain the vertical is described well in the model. The RVMR is only 20%-60% of the surface value, and depending on the NH3 profile doesn't have to be in relation to the surface. Some words on any effects caused by the RVMR and maybe a comparison on the profile of TES and the model? Similar to the study by shephard et al?: Shephard, M. W., McLinden, C. A., Cady-Pereira, K. E., Luo, M., Moussa, S. G., Leithead, A., Liggio, J., Staebler, R. M., Akingunola, A., Makar, P., Lehr, P., Zhang, J., Henze, D. K., Millet, D. B., Bash, J. O., Zhu, L., Wells, K. C., Capps, S. L., Chaliyakunnel, S., Gordon, M., Hayden, K., Brook, J. R., Wolde, M., and Li, S.-M.: Tropospheric Emission Spectrometer (TES) satellite observations of ammonia, methanol, formic acid, and carbon monoxide over the Canadian oil sands: validation and model evaluation, Atmos. Meas. Tech., 8, 5189-5211, doi:10.5194/amt-8-5189-2015, 2015. Also the observed concentrations near Bakersfield seem to be quite low at time. Any effects due to the sensitivity/retrieval of TES for these low retrieved concentrations? You could add a figure with the observed and modelled profiles and the AVK of the satellite observations to show the difference in the vertical (and yes the DOF are low but the profiles are still used for the RVMR).*

We have added an additional figure (Figure 8) that follows a similar evaluation performed in Shephard et al. (2015, now included as a reference) and have included additional discussion of the satellite analysis (Section 4.3). The bias in the $CMAQ_B$ and $CMAQ_{AB}$ results relative to the TES RVMR has also been added as Table 3. The added discussion based on the new Figure 8 is as follows (Page 13, lines 25-33):

> "However, the model RVMR can be very sensitive to errors in the modelled vertical distribution of $NH_3$. We investigated this by comparing each level of the TES retrieved $NH_3$ profile with the corresponding CMAQ profile level after the observation operator is applied. Figure 8 shows box-and-whisker plots of this comparison for the $CMAQ_{base}$ and $CMAQ_{AB}$ model scenarios. This plot differs from that in Shephard et al. (2015) in that it includes the average of layers below 908 mb, which introduce an RVMR bias due to levels that are below 1000 mb. For $CMAQ_{base}$, there is a substantial negative bias in the lowest level (-5 ppb), but for $CMAQ_{AB}$ this switches to a positive, smaller bias (~1 ppb). Furthermore, the other, higher levels show little bias (~0.08 ppb). Thus comparing the TES and CMAQ profiles level-by-level indicates that the $CMAQ_{AB}$ scenario performs the best in simulating the TES retrievals, consistent with the conclusions based on the surface observations in Section 4.1."

*6) Aircraft observations: Possible artifacts? Include the uncertainties in the discussion of the results / conclusions.*

We have added Page 5, lines 25-30 to better explain artifacts and uncertainties from aircraft measurements using the CIMS instrument and added an additional reference (Nowak et al., 2007) and point to these uncertainties in the Discussion Section (Page 14, line 4). The new lines are as follows:

> "The CIMS instrument sampled air through a 0.55 m long heated teflon inlet with a fast flow. Measurement artifacts were accounted for by quantifying and subtracting the background signal originating from $NH_3$ desorption from instrument surfaces. The background signal was determined in flight by actuating a teflon valve at the inlet tip once every half hour to divert the sample air through a scrubber that removes $NH_3$ from the ambient air stream (Nowak et

al, 2007).  Additionally, standard addition calibrations from a $NH_3$ permeation
tube were performed several times each flight to determine instrument
sensitivity."

*7). Model: Some discussion on performance of the model for the vertical distribution of NH3
would be helpful.*

We have provided an additional flight curtain plot showing changes in vertically modeled $NH_x$ for
the $CMAQ_B$ run (Figure 5c) as well as added discussion of these results aloft (Page 12, lines 6-11).
Table 2 also includes statistics for the $CMAQ_{AB}$ runs as compared to flight measurements. In
general we see an increase aloft in $NH_x$ from increased net land-atmosphere flux from the
bidirectional $NH_3$ calculations.

> "Figure 5c is consistent with these results (and inconsistent with the hypothesis
> that vertical mixing is underestimated in the model) as the vertically distributed
> concentration of $NH_x$ significantly increases from the $CMAQ_{base}$ case to the
> $CMAQ_B$ case. The transport of $NH_3$ also tends to increase, this being a potential
> explanation for the plume entering the plot domain around 21:00 PDT in the
> bottom curtain plot. The total column concentration of $NH_3$ also increases,
> leading to a significant positive model bias for the $CMAQ_{AB}$ and $CMAQ_B$
> scenarios (e.g. in the earlier part of the flight in Figure 5c and Table 2),
> suggesting a possible overestimation of total $NH_x$ emissions by the bi-directional
> $NH_3$ scheme during the afternoon and evening hours that the flights took place."

We have also included an additional figure (Figure 8, below) that follows a similar evaluation
performed in Shephard et al. (2015) and is described in comment response # 5.

[Figure]

Figure 8. Boxplots of a) TES $NH_3$ retrieval by pressure level, b) TES $NH_3$ retrieval averaging
kernel (AK) diagonal, c) difference between the TES $NH_3$ retrieval and $CMAQ_{base}$ modelled $NH_3$
interpolated to TES levels with an AK applied for the baseline model run and d) same as c) but for

the CMAQ$_{AB}$ run. Box plots show the mean (green), median (red), interquartile range (IQR, blue box), whiskers at 1.5 IQR and outliers beyond that.

*8) Also include some words on the performance for species like HNO3 and sulfates as these are probably causes for any discrepancies in the diurnal cycle.*

We have included a ground measurement analysis plot for HNO$_3$ and SO$_4$ in the supplement (Figure S3), but we note that our method of primarily looking at NH$_x$ in our aircraft and surface evaluations removes potential errors due to incorrect gas-particle partitioning of NH$_3$, and that gas-phase NH$_3$ dominates the NH$_x$ concentration in this region. Thus we have not provided and evaluation of modeled HNO$_3$ and SO$_4$ in the main paper. The discussion added to the paper regarding these results is on Page 12, lines 10-16 and quoted below.

> "As noted above, the results for NH$_{3(g)}$ generally track the results for NH$_x$ already discussed. In contrast, the model usually under-predicts the small amount of NH$_{4(p)}$ observed (on average < 1 ppbv, Figure 2c) by a factor of 2, with little variation between the model scenarios (Table 1). These model errors in NH$_{4(p)}$ reflect not only model errors in total NH$_x$, but also errors in the formation of HNO$_{3(g)}$ and SO$_{4(p)}$ (Figure S3). HNO$_{3(g)}$ is overestimated in all model simulations up to a factor of 4, with concentrations not changing between model cases. SO$_{4(p)}$ measured concentrations are minimal and don't appear to have any trend and also do not change with model cases. However, as our interest in this study is in constraining NH$_3$ emissions, not inorganic aerosol formation, we do not investigate these errors further here."

[Figure]

Figure S3. The CalNex ground measurements at the Bakersfield site (solid black) compared to the CMAQ$_{base}$ (solid blue), CMAQ$_{AB}$ (purple) and CMAQ$_B$ (green) simulations for a month of model runs. The top panel (a) shows SO$_{4(p)}$, b) shows HNO$_{3(g)}$

*9) 2.3. PBL: Figure 7 shows the performance of the WRF PBL when compared to the HSRL observations. The authors conclude that the deviations are not a probably cause for any faults in the diurnal cycle of NH3. I do not agree with this conclusion. From the plot one can conclude that*

*for small PBL heights there are large deviations up to a factor 2 when compared to the modelled WRF PBL. You can convince me by showing a figure of the diurnal cycle of the PBL for both HSRL and WRF? And/or the normalized version of the cycle? By adding a diurnal cycle figure you also strengthen any comments and conclusions in the paper that the errors in the PBL have no effect.*

We have replaced the scatter plot with a time series of HSRL and WRF PBL (Figure 4 in the revised paper) in order to identify any diurnal patterns. The results from the scatter plot remain in the text for reference and in the Supplement Material (Figure S4). Although flight measurements only last a few hours and the diurnal cycle cannot be determined from these measurement comparisons, the plot does show that for small PBL heights the deviations are no greater than for higher heights and there do not appear to be any biases that could significantly contribute to the diurnal cycle of $NH_3$.

*10) 2.x Emissions: Can you add a table or a short paragraph on the emissions sources and their relative totals?*

We have added a table to the supplement (Table S2) that describes the fraction of CARB $NH_3$ emissions by source for the counties in the San Joaquin Valley and added a discussion on how this may impact the model results (Page 11, lines 19-22).

> "For Kern County, where Bakersfield, CA resides, pesticide/fertilizer applications dominate the $NH_3$ emissions inventory at 72%, followed by farming operations at 25%, and other sources for the remaining fraction. Table S2 in the Supplemental Material describes the fraction of $NH_3$ emissions for counties in the SJV."

*11) 3.3 Hysplit, I think this section can be removed as in the remainder of the paper only 3 sentences are dedicated to the results.*

We agree with the reviewer and have removed the HYSPLIT figure from the paper and added it to the supplemental material (Figure S2). However, we kept the discussion of the HYSPLIT model results as we feel addressing the direction from which emissions may have come is important in understanding the $NH_3$ cycle in the area.

*12) 4. Results: A bit of extra structure and discussion in the results will greatly improve the manuscript.*

We agree with the reviewer and have re-arranged sections so as to organize the paper around the three measurement platforms (Section 4.1 surface, Section 4.2 flight and Section 4.3 satellite) a new Discussion Section (Section 5) and an improved Conclusion Section (Section 6), as described in the introduction to the model evaluation section and earlier in this reviewer response. The response to comment 14 below describes the new Discussion Section in more detail.

*13) The authors have a wealth of data available but only scarcely use it. The flight data is only used for the basic version of the model, and not discussed in the latter parts of the manuscript, while the variation in the emissions will also affect the vertical distribution of the $NH_3$ concentrations.*

In addition to expanding our model sensitivity evaluations to a full month, we have also extensively added to the model comparisons to flight data. This includes a much more in-depth table (Table 2) which now includes flight data comparison to all model runs, as well as an added figure (Figure 5c) that demonstrates the changes in the vertical profile of $NH_x$ when the bi-directional ammonia scheme is applied. In general model concentrations increase in both the surface layer and aloft due to an increase in the net land-atmosphere flux, which can be interpreted as a combination of decreased deposition and increased emissions. However, this also increases

the mean bias considerably (+7-10 ppbv) as seen in Table 2, and discussed on Page 12, lines 28-33 and Page 13, lines 1-3.

> "The $CMAQ_{base}$ run does not take this into consideration, but when the bidirectional flux exchange of $NH_3$ is calculated in $CMAQ_{AB}$ and $CMAQ_B$, $NH_3$ dry deposition should generally decrease, increasing the net land-atmosphere flux (Bash et al., 2013). Figure 5c is consistent with these results (and inconsistent with the hypothesis that vertical mixing is underestimated in the model) as the vertically distributed concentration of $NH_x$ significantly increases from the $CMAQ_{base}$ case to the $CMAQ_B$ case. The transport of $NH_3$ also tends to increase, this being a potential explanation for the plume entering the plot domain around 21:00 PDT in the bottom curtain plot. The total column concentration of $NH_3$ also increases, leading to a significant positive model bias for the $CMAQ_{AB}$ and $CMAQ_B$ scenarios (e.g. in the earlier part of the flight in Figure 5c and Table 2), suggesting a possible overestimation of total $NH_x$ emissions by the bi-directional $NH_3$ scheme during the afternoon and evening hours that the flights took place."

*14) The systematic discussion of possible causes for the discrepancies between the modelled and measured concentrations as given in 4.2 should be added for the other versions of the model. Each version should rule out one or more of the possible causes, which will add to the overall discussion of the state of the model (and not just this model, but the overall performance of most CTMs).*

As noted above, we have significantly expanded our discussion of the discrepancies between the modeled and measured concentrations for al three scenarios in Section 4, and have added a new Discussion Section (Section 5) that discusses the remaining errors in the $CMAQ_{AB}$ scenario and suggests avenues for further model improvement. We find that all three datasets suggest that the remaining errors in modelled $NH_x$ concentrations in $CMAQ_{AB}$ are due to the diurnal profile of the net land-atmosphere $NH_3$ flux in the $CMAQ_{AB}$ run peaking too late in the day or due to errors in the dynamic emissions response of the bi-directional $NH_3$ scheme to local temperature and wind speed conditions (Bash et al., 2013). For example, this could be due to errors in the dependence of soil conditions (e.g., soil temperature, pH, and water content) on meteorology and crop management practices as calculated within the bi-directional $NH_3$ scheme (Cooter et al., 2012). Additionally, aircraft results may also suggest errors in the vertical mixing of $NH_x$ during the afternoon and evening (e.g., the peak of the PBL height and the collapse). While we consider this effect as likely less important to the remaining errors in $CMAQ_{AB}$ than the potential errors in the bi-directional $NH_3$ scheme, an overestimate of vertical mixing during the afternoon would overestimate the flux of $NH_x$ from the surface layer of the atmosphere to the upper levels, consistent with the aircraft overestimate. In addition, the soil-canopy-surface atmosphere system would respond to this overestimate of vertical mixing by increasing the net flux of $NH_x$ from the soil to the atmosphere in order to maintain equilibrium, resulting in a total overestimate of the emissions of $NH_x$ during the afternoon and evening.

*15) Adding a table with the airborne observations vs each of the modelled versions would help. Table 2. add some correlations and statistics similar to table 1.*

As described above, we have included more detailed tables (Table 1 - ground measurements, Table 2 - flight measurements and Table 3 - satellite comparison) that describe the statistics for all model scenarios.

*16) A figure or table in which you split the statistics per hour of the day will give some further insight on the performance of the model for each part of the day. Partially this is done already in figure 4, but some correlations / bias plot could be added for more information. In this figure/table one can then easily point out the improvements in the later model versions similar to figure 8.*

We have kept the flight measurement comparison for May 24, 2010 split up by time as well as Figure 3 which contains the average hourly ratio of $CMAQ_{base}$ modelled to measured $NH_3$ and $NH_x$ mixing ratios and the average $CMAQ_{base}$ modelled RVMR to TES RVMR ratio, a boxplot of average hourly $CMAQ_{base}$ modelled and measured $NH_x$ mixing ratios for the Bakersfield ground site, averaged over all measurement days during CalNex. We feel that the new Figure 9, similar to Figure 3 but for $CMAQ_{AB}$ results, demonstrates results that contain a model bias in the afternoon-evening hours, with slightly higher concentrations in the afternoon peaking around 19:00 (Figure 9b). We attribute these remaining errors to a few possible reasons discussed in comment # 14 and in more detail in the new Discussion Section 5.

*17) 5. Conclusion / Discussion: I am missing a final discussion on how one would improve the model in the future or what kind of measurements would be needed (does not have to be long). A few points for a start of the overall discussion and state of CMAQ/NH3 modelling (bit broad): What kind of measurements would the authors perform to further understand the model and reasons for discrepancies between modelled and observed concentrations – Discuss point by point what this study improved, for now I can only see a small improvement to the bias. -- Final words: I recommend rewriting some parts of the manuscript following a few of the stated highlights to improve the overall quality of the paper. When rewritten this paper can be a great start for future model (improvement) studies.*

We thank the reviewer for their helpful comments regarding suggestions for a better organization of the paper and their technical comments. We feel that with our new organization of the Results, Discussion, and Conclusions section, described above, and the additional technical improvements made as suggested by the reviewer, the revised manuscript has been improved to better communicate both the model results and our recommendations about future steps towards future model improvements.

**Reviewer #2**

*1) General comments: This study presents a study combining surface, aircraft and satellite
measurements of NH3 and NH4 concentrations in the San Joaquin Valley with a model study
using CMAQ. The approach taken enables the authors to identify lacks in knowledge in both
model description and emission inventories. While this is a worthwhile effort, the analysis and
discussion could be improved by more explicitly including a discussion section in which the
possible explanations of mismatch between model and observation are listed, as well as an
outlook section with possible improvements to the model or emission data. If these points are
improved upon (I give a few suggestions below), this paper could really contribute to improving
NH3 modelling and to a better understanding of the sources of mismatch between model and
measurements.*

We agree with the reviewer's comment that the paper could be improved by adding an
explicit Discussion Section as well as a discussion of possible future model and inventory
improvements. We have addressed this by, first, rearranging the model results section (Section 4)
to organize the paper around the three measurement platforms (ground, flight and satellite
measurements) rather than around the different model sensitivity studies. Section 4.1 in the new
manuscript evaluates a full month of output from the three different model scenarios, $CMAQ_{base}$,
$CMAQ_B$ (bidirectional ammonia) and $CMAQ_{AB}$ (bidirectional and adjusted emissions), with the
surface measurements. We thoroughly discuss the evaluation of the diurnal distributions of $NH_x$,
$NH_{3(g)}$ and $NH_{4(p)}$ in the three scenarios, since the surface measurements provide the best available
information for identifying diurnal patterns. Section 4.2 discusses the aircraft measurement
comparisons as well as the vertical profile of $NH_x$, since the aircraft measurements provide insight
into the vertical dispersion of $NH_x$ as well as the overall magnitude of emissions in the afternoon
and evening hours. Section 4.3 discusses the comparison of the three scenarios with the TES
retrievals, including an added level-by-level comparison of the modeled and retrieved vertical
profiles of $NH_3$. The new Discussion Section (Section 5) then describes the remaining errors in the
final model version ($CMAQ_{AB}$) and suggests possible explanations for these errors and directions
for future research. The revised Conclusions Section (Section 6) summarizes the results and
discussion, and now includes a description of the overall model bias changes as well as
suggestions for future work on $NH_3$ modeling in the SJV.

The new Discussion Section states that all three datasets suggest that the remaining errors in
modelled $NH_x$ concentrations are due to the diurnal profile of the net land-atmosphere $NH_3$ flux in
the $CMAQ_{AB}$ run peaking too late in the day, possibly due to errors in the dynamic emissions
response of the bi-directional $NH_3$ scheme to local temperature and wind speed conditions (Bash
et al., 2013). For example, this could be due to errors in the dependence of soil conditions (e.g.,
soil temperature, pH, and water content) on meteorology and crop management practices as
calculated within the bi-directional $NH_3$ scheme (Cooter et al., 2012). Additionally, the aircraft
results also suggest errors in the vertical mixing of $NH_x$ during the afternoon and evening (e.g., the
peak of the PBL height and the collapse). While we consider this effect as likely less important to
the remaining errors in $CMAQ_{AB}$ than the potential errors in the bi-directional $NH_3$ scheme, an
overestimate of vertical mixing during the afternoon would overestimate the flux of $NH_x$ from the
surface layer of the atmosphere to the upper levels, consistent with the aircraft overestimate. In
addition, the soil-canopy-surface atmosphere system would respond to this overestimate of
vertical mixing by increasing the net flux of $NH_x$ from the soil to the atmosphere in order to
maintain equilibrium, resulting in a total overestimate of the emissions of $NH_x$ during the
afternoon and evening.

We recommend that future modelling work includes updating the CARB $NH_3$ inventory to
account for $NH_3$ from fertilizer, livestock, and other farming practices separately, as well as
adding information on crop management practices specific to the SJV region to the EPIC-FESTC
system. We also recommend top-down studies that focus not just on correcting the net $NH_x$ flux to
the atmosphere but also on determining the diurnally-varying biases in the canopy compensation
point that determines these net fluxes.

*2) Specific comments: For readers not familiar with the SJV geography and the location of the Bakersfield site, providing a map of the region could be valuable.*

We agree and have provided a new Figure 1 in the paper and below that shows the geography in California's SJV as it relates to $NH_3$ emissions. In addition, this plot shows the location of the three measurement platforms used, in a map display. We have also provided a Table S2 (in the Supplement and below) that details the fraction of $NH_3$ emissions from different area sources in the SJV.

[Figure]

Figure 1. Distribution of $NH_3$ emissions across California (background) on May 12, 2010 at 19:00 UTC as well as P3 flight tracks (small circles), TES transect (green squares), and the Bakersfield site (red star)

Table S2. Contribution of sources to $NH_3$ emissions inventory in the San Joaquin Valley as reported in the CARB emissions inventory.

| County | Pesticide/ Fertilizer Fraction | Farming Operation Fraction | Other Area Sources |
|---|---|---|---|
| Kings County | 0.47 | 0.55 | 0.00 |
| Fresno County | 0.40 | 0.57 | 0.03 |
| Kern County | 0.72 | 0.25 | 0.03 |
| Merced County | 0.23 | 0.76 | 0.01 |
| Stanislaus County | 0.32 | 0.65 | 0.03 |
| Madera County | 0.33 | 0.64 | 0.03 |
| San Luis Obispo County | 0.25 | 0.51 | 0.24 |
| Tulare County | 0.11 | 0.86 | 0.02 |

*3) Title: Not everyone is familiar with CalNex. Adding 'campaign' (or otherwise clarifying the term) at the end of the title would make it clearer.*

We agree with the reviewer and have added 'campaign' to the title that now reads "Modeling the Diurnal Variability of Agricultural Ammonia in Bakersfield, California during the CalNex Campaign".

*4) Introduction: While the introduction presents a thorough overview of previous work, it is rather elaborate. Condensing this by focusing on the most important points would increase readability. This is later also true for the description of the TES data and the CMAQ model.*

We agree with the reviewer's comment and have made an effort throughout the paper to condense discussion, and also feel with the improvements made based on both reviewer's comments that the overall paper readability has been improved. For example, we have removed paragraphs in the Introduction Section (Page 3, lines 25-27 in the original manuscript) and Section 3.2 (Page 8, lines 20-25 in the original manuscript) that we felt were not necessary for background understanding of the manuscript.

*5) Please specify at some point in which period the CalNex campaign was active.*

The CalNex campaign ran in May and June of 2010 as stated in the paper on Page 4, line 15.

*6) Data: Please give coordinates and elevation of the Bakersfield site.*

We have provided coordinates and elevation of the Bakersfield surface site, Page 6, line 5 – "Bakersfield, California is located on the southern part of the SJV (35.35°N, 118.97°W, 20 m asl)"

*7) Models: In the text on the CMAQ model results of sensitivity studies are already provided. Consider moving this to the results section. You could also consider dedicating a paragraph to the description of the emission database (which are the most important ammonia sources, etc.) as this is so important in your uncertainty analysis later.*

We have made sure no results are described in sections previous to Section 4 and have added a discussion on the $NH_3$ emissions database (Page 11, lines 19-22) to the paper:

> "For Kern County, where Bakersfield, CA resides, pesticide/fertilizer applications dominate the $NH_3$ emissions inventory at 72%, followed by farming operations at 25%, and other sources for the remaining fraction. Table S2 in the Supplemental Material describes the fraction of $NH_3$ emissions for counties in the SJV."

We have also improved Figure 1 to demonstrate the spatial distribution of $NH_3$ sources in the SJV and have included a table that describes the break down of area ammonia sources in the Supplement (Table S2), as described in comment 2.

*8) Page 8, line 32: ' . . . soil emissions potential and NH4': sentence is incomplete.*
We have restructured and completed this description sentence now on Page 9, lines 19-21:

> "Finally, we also ran CMAQv5.0.2 using the bi-directional $NH_3$ flux scheme as developed by Bash et al. (2013) that uses fertilizer application data, crop type, soil type, and meteorology from MCIP output to calculate soil emissions potential and $NH_4$ to simultaneously calculate $NH_3$ deposition and emission fluxes for the CMAQ US domain."

*9) Results:*

*Section 4.1: You claim that the relative changes in NH3 concentration along the transect are captured well by the model, but to me this seems not to be the case: the highest concentrations are underestimated much more strongly, also in a relative sense, than the lower concentrations outside the direct source region. Also, based on figure 3 you conclude that CMAQ with the CARB inventory captures the spatial variability near Bakersfield well, but given the correlation coefficient of 0.22 for the overpasses closest to Bakersfield I'm not sure this statement holds. The purpose of highlighting these points is not clear as it is later not at all discussed. If I understand the plot and caption correctly, each point represents one overpass in one grid cell, i.e., this plot shows both temporal and geographical variability. Is this correct? If yes, could you comment on which part of the scatter is caused by temporal and which by geographical variability?*

We have clarified the discussion of model results compared to the TES overpass in Section 4.3 and improved the wording on the relative correlation comparisons. Figure 2 from the original paper (now Figure 6 in the revised paper) demonstrates that the $CMAQ_{RVMR}$ is consistent with the location of higher concentrations of $NH_3$ as seen in the $TES_{RVMR}$. We have also clarified our comments on the temporal/geographical variability – in general there is no actual temporal variability in the TES-CMAQ plots, since CMAQ output is hourly and only the hour corresponding to the TES overpass is shown.Thus there is only geographical variability. The description now reads (added, Page 13, lines 5-10):

> "Figure 6a shows the RVMR retrieved from the TES spectra ($TES_{RVMR}$) for one overpass (during one hour of model output) on 12 May 2010; the other overpasses during the campaign are similar. Figure 6b shows the equivalent $CMAQ_{base}$ modelled $NH_3$ RVMR ($CMAQ_{RVMR}$) (see Equation 1 and 2 in Section 2.2), and Figure 6c shows the difference between the two. This figure demonstrates that the $CMAQ_{base}$ case can identify the locations of different sources of $NH_3$ and the resulting geographical relative changes in $NH_3$ along the transect, but that the $NH_3$ RVMRs are underestimated, particularly at higher $NH_3$ RVMRs (Table 3 and Table S2)."

The graph below shows the average TES RVMR (ppb) for each overpass day during the CalNex campaign. It can be seen that no additional temporal trends could be discerned (i.e. on a month-to-month basis) and thus this potential temporal variability was not discussed in the paper.

| Date | Average TES RVMR (ppb) | Std Dev. TES RVMR (ppb) |
|---|---|---|
| 2010/05/12 | 8.59 | 7.99 |
| 2010/05/14 | 6.54 | 6.00 |
| 2010/05/28 | 4.55 | 2.99 |
| 2010/05/30 | 6.33 | 6.19 |
| 2010/06/13 | 8.26 | 5.69 |
| 2010/16/15 | 8.47 | 5.53 |

*10) Section 4.2: From figure 4 I don't see an underestimation of a factor 2.5 during the daytime, rather 1.5-2.*

We thank the reviewer and realize how this could be interpreted as 1.5 to 2 and have corrected this on Page 10, lines 13-19 to now say:

> "The model bias shows a clear diurnal cycle, with $CMAQ_{base}$ significantly overestimating surface $NH_x$ concentrations at night by up to a factor of 4.5 and generally underestimating $NH_x$ during the daytime at 0.6 between 13:00 and 14:00 local time, consistent with the average $TES_{RVMR}$ observations near Bakersfield at about 13:30 local solar time plotted as the green dot in Figure 3a and further discussed in Section 4.3. These results suggest that the constant daily emissions for agricultural $NH_3$ emissions in the CARB inventory (blue line, Figure S1 in the Supplemental Material) may be misrepresenting the diurnal

emission patterns suggested by the measurements. This is consistent with previous work done in North Carolina; Wu et al. (2008) found that $NH_3$ emissions from livestock feed lots show a strong diurnal cycle, peaking at mid-day.

*11) Line 26-27 (page 9) would be better supported by adding a time series of the measurement-model comparison to show the seasonal patterns.*

We agree with the reviewer and have added a new Figure 2, that now includes a full month comparing model results to ground measurements for $NH_x$, $NH_{3(g)}$, $NH_{4(p)}$, temperature, and wind speed. The temporal scope of these measurements, however, is only May and June so longer seasonal patterns cannot be evaluated with this dataset.

*12) Lines 3-6 (Page 10) seem redundant.*

We agree with the reviewer and have removed this section.

*13) Lines 13 and onwards (page 10): please mention that you now compare concentrations at 400+ meters above the surface; otherwise the step from the ground-based observations to air craft might be confusing.*

We have re-worded the sentence (now on Page 12, line 10) and elsewhere to include the fact that these measurements are taken at higher altitudes, as shown in Figure 5.

> "The aircraft observations in the SJV indicate a large underestimate (range of factors about 1 to 5) in $CMAQ_{base}$ modelled $NH_x$ concentrations at higher altitudes as shown in Table 2 (all flights in SJV) and Figure 5 (two flights)."

*14) Page 10, line 28 – page 11, line 7: This section can be shortened significantly; consider if results that are not worth showing are worth talking about.*

With the re-arranging of the model results section, and the addition of the new Discussion Section we feel that all results presented now are worth discussion. We have also excluded the plot of a brief HYSPLIT analysis that is only worth mentioning to rule out significant diurnal changes in source region that potentially could have contributed to biases in modeled concentrations, but do not appear to.

*15) Section 4.3: What does 'consistent with measured temperature patterns' mean? I assume it suggests that temperature is the driving variable for the emission variability during the day, could you state that more clearly?*

We realize the initial reference to temperature patterns was confusing. We now include model and measured temperature at the Bakersfield site, Figure 2d and discuss how temperature variability could influence modeled concentrations (Page 14, lines 27-33, and Page 15, lines 1-4):

> "Thus the remaining errors are less likely related to errors in atmospheric meteorological conditions, and are more likely due to errors in the dependence of soil conditions (e.g., soil temperature, pH, and water content) on meteorology and crop management practices as calculated within the bi-directional $NH_3$ scheme (Cooter et al., 2012). The scheme calculation assumes two soil layers (0.01 m and 0.05 m) that independently exchange $NH_x$ with the canopy, which then exchanges $NH_x$ with the surface layer of the atmosphere (Bash et al., 2013). If the calculation of the response of soil properties in these layers to surface meteorology and crop management practices is incorrect (e.g., the soil layers do not heat up or cool down quickly enough with the change in surface temperature), that would affect the amount of $NH_x$ available from the soil as

well as the rate at which the soil $NH_4^+$ is converted to $NO_3^-$ through nitrification (Bash et al., 2013). This would result in errors in the flux of $NH_x$ from the soil to the canopy, thus altering the canopy compensation point and the net atmospheric flux."

*16) Why did you only adjust the hourly emission profile for NH3, was there no day-to-day variability (e.g. related to temperature) to take into account? With the approach taken, you assume that concentrations in a certain hour are dominated by emissions in that same hour; could you comment on this assumption?*

We focused on the diurnal errors, as these errors can complicate the interpretation of data sets, such as the aircraft and TES observations, that do not cover the entire diurnal cycle. While temperature certainly varies day-to-day, and this should affect $NH_3$ emissions, we are less confident that there is sufficient variability in temperature during our one-month evaluation period to constrain this variability. Other varying factors, such as source location, planetary boundary layer height, and gas-to particle partitioning, are discussed and do not show significant enough biases to explain modeled $NH_3$ concentration bias.

The model appears to capture the day-to-day meteorological variability well, as seen in Figure 2d, which shows the modelled and measured surface temperature and wind speed. We do feel, however, that errors in the dependence of soil conditions (e.g., soil temperature, pH, and water content) on meteorology and crop management practices as calculated within CMAQ may be contributing to the modelled $NH_3$ bias. Additionally, the aircraft results also suggest errors in the vertical mixing of $NH_x$ during the afternoon and evening and while we consider this effect as contributing less to the remaining errors in the model, an overestimate of vertical mixing during the afternoon would overestimate the flux of $NH_x$ from the surface layer of the atmosphere to the upper levels. Thus, we recommend that future work to improve the simulation of atmospheric $NH_x$ concentrations in the SJV focus on bottom-up and top-down approaches that will better estimate the diurnal changes in the canopy compensation point that determines the net flux from the land to the atmosphere in the bi-directional $NH_3$ scheme (Bash et al., 2013). This is discussed in more detail in the new Discussion Section 5.

*17) Why did you decide to test the new diurnal profile for 7 days only? A comparison to the aircraft data would be valuable here as well, to see to what extent the changed diurnal profile impacts modelled concentrations and model performance at higher altitudes.*

As noted previously, we have extended the time series of all model runs to cover an entire month (May 22 – June 22) which covers the complete Bakersfield super site record for $NH_x$. Figure 2 compares our three model scenarios to the ground measurements for $NH_x$, $NH_3$, and $NH_4$ for this entire period, and the statistics for this period are shown in the new Table 1. In addition, we have added an additional plot of aircraft data compared to the $CMAQ_B$ run (Figure 5c), which shows the impact that the calculation of bi-directional exchange of ammonia has on the model performance at higher altitudes. The statistics for these flight comparisons are also shown in Table 2.

*18) Section 5: This section would be stronger if it contained more than a summary of the most important points of the paper, but also a discussion on future steps / important work to be done to improve the modelling of ammonia and the representation of emissions. For example, a discussion on the relative importance of the misrepresentation of emission diurnal cycles vs. misrepresentation of the vertical mixing (which should we work on first?) would be valuable.*

We thank the reviewer for this valuable recommendation and feel that the new Discussion Section, described in comment # 16 above, emphasizes the relative importance of several aspects that could improve the diurnal modeling of ammonia.

*19) Technical comments:*
*Page 2, line 6: photoxidize should be photo-oxidize*

We have corrected this typo.

*20) Page 3, line 11: CONUS might not be a known acronym for non-US readers; please explain.*

We have removed CONUS and replaced with 'continental US'

*21) Page 3, line 25: Write out TES as it is the first mention in the main body of the article. In general: check for unexplained abbreviations.*

We have added this and checked the paper for other abbreviations not defined.

*22) Page 7, lines 11 and 12: HSRL is mentioned as acronym but only written fully at the second instance.*

We have corrected this by adding the acronym description the first time it is mentioned, Page 4, line 27.

*23) Page 8, line 14: SoCAB is not explained.*

We thank the reviewer for pointing out this referenced term in the paper. This sentence and the one following it describe emissions in the southern and LA areas, unrelated to the emissions in the SJV, thus we removed these 2 sentences.

*24) Page 9, line 13: scatterplot should be scatter plot*

We have corrected this typo

*25) Page 11, line 23: remove 'mostly' as the CARB NH3 emissions are completely constant.*

We have removed 'mostly' from the description of the daily emissions pattern.

---

## Editor Decision (ED1)

**Suggestions for revision**:

The recent revision has greatly improved the manuscript, and in my opinion could be improved to be ready for publication after a final set of revisions. The requested revisions mostly cover the results & discussion as to my opinion some conclusions are made to lightly and/or are not completely supported by the results the authors show. A few comments/edits are minor details and will only take a few minutes to correct. A few others are more in depth and touch the basis of the manuscript. The authors do not need to completely agree to all statements but some elaboration will be needed to convince me and to my expectation most of the readers.

Minor edits:

Table 1; Add a description of MB and MNB, also add this to the text. Add the number of observations?

Discussion of Table 1;; you show correlations of around 0.001 to 0.05. At this point statistics about slopes and MB become more or less irrelevant as you are applying fits to clouds of scatter. Maybe some explanation as to why the correlations are so low, at least mention it in the text. Correlations are almost not mentioned in the text, unless they are high ~0.7…, don't hide the fact that the model misrepresents the measured values even after the CMAQab additions.

Table 2; similarly to table 1, add a description of MB and MNB. Add the number of observations?

See the in depth comments for an explanation/question on the effect of wind speed in combination with wind direction and possible sources.

Table 3; add description of MB and MNB. Add the number of observations?

Figure 1; Colorbar label: Add an E to mission. Also change font to the same font of the colorbar ticks?

Figure 2; If possible add Wind Direction, My explanation is added in the in depth comments.

Figure 3; If possible make one big figure with 4 subplots using figure 3a b and 9 a b. This will make it easier for the reader to compare the old and new situations,

Also add the Blue "observed" plot to 9b.

Figure 4, Good figure, as for colors, maybe blue and red? Easier to distinguish the differences.

Figure 5, Maybe also add 2010/06/16 CMAQ B for the top figure? Else remove the top subplot. What about CMAQ AB? Another possibility: Change it to 2 figures, one for 2010/06/16 and one for 2016/05/24 both with 3 subplots, CMAQ base, B and AB. Also show the figures in chronological order.

Figure 7, Add in CMAQ B, CMAQ AB for comparison.

Page 7, line 28; W is a weighting matrix, add a few words on how the matrix is calculated (possible effects following such a mapping).

In depth comments/discussion:

**Overall** & page 11 line 12-20 about the scaling factor of the emissions; You describe that you directly scale emissions needed to match a ratio of measured to modelled concentrations. What you more or less assume in this case is that the ammonia emissions is linearly related to ammonia concentration.

Can you shortly discuss the scientific basis behind this assumption, and why not use a somewhat lower or constant factor?

**Page 10, line 10-14**, you mention the effect of hourly varying emissions, and that the diurnal variation is missing. What about the monthly variation of the different sources?

**Overall & discussion of effects of transport;** I do not fully agree with the explanation / conclusion that transport of ammonia does not seem to be a major factor. Although you cover the basis of wind direction there is a short discussion missing on the effects of wind speed. Towards the north-west the ratio between livestock/fertilizer applications is probably different compared to the local conditions at Bakersfield, CA. As my personal knowledge of the counties surrounding Bakersfield is non-existing I cannot couple the summary of the sources given in S2 to the concentrations measured at the Bakersfield site (when combining this to the wind speeds, and assuming a more or less constant wind direction.  Would it be possible to add a figure in which you compare windplots of the model (a) with measured concentrations (b). Radially you can show the measured concentrations, coloring can show the wind speed (or vice versa). This will support the explanation that transport / horizontal misrepresentation of the emissions are not a major cause of the difference between model and measured concentrations. To somewhat support my statement; you show that the model underestimates the RVMR to the north west compared to the satellite, while locally its basically the same, wind speeds increase to the end of the day compared to the overpass time of TES (13:30). Even if you would perfectly model the local emissions in the Bakersfield model cells, the misrepresentation to the north-west could possibly cause of bias shown in figure 3 & 9.

Table S2; Add some coordinates or relative position compared to Bakersfield. Or a small map showing the counties? Also what does Farming Operation mean? And what about livestock? Is this part of Farming Operation?

**Overall discussion of the statistics in Table 1 & 2;** What I am missing is a in depth discussion of the statistics given in Tabel 1 and 2. What essentially is shown in Table 1 is that there is zero correlation between the model and the measurements, i.e. any statements made on the bias will not convince anyone and I think it does not fully reflect the performance of the CMAQ model as even using CMAQab only gives a correlation of 0.05. A few lines on why these correlations are so low will improve the manuscript and re-establish faith in the CMAQ models capabilities to simulate NH3.

Table 2 & Section 4.2; At a first glance I would conclude that CMAQ base is better than CMAQab or CMAQb. Even though the diurnal variation is improved for the hours between 1 AM and 6 AM, the overall levels for the other ¾ of the day are still too high or too low depending on the hour. Only a short discussion is given of CMAQB and CMAQab, page 12, line 24-31. You correctly point out that the emissions are now far too high for most of the day for both "improvements". The same is visible in Table 1 and somewhat less in table 3 as TES only gives a snapshot of the situation at 13:00.

This brings me to a point to question the value of CMAQb and CMAQab to the manuscript. While the authors do a good job describing the possible causes of error in the base model, the new additions do not improve the model and are thus somewhat irrelevant in the current state besides showing that it is not correct to scale emissions following concentrations and that the current bi-directional schemes are far from perfect, things that have been shown before. I would like to put forward two possible approaches to make the manuscript ready for publication.

1. Change the manuscript to fully focus on the performance of CMAQbase & CMAQb. These two versions of the model have been described before in earlier publications and only a small addition to the result section, to better cover the CMAQ b results, will be needed to

improve the manuscript enough for publication. Especially as possible causes of CMAQb are already mentioned in the discussion.

2. If the authors want to keep in the CMAQb model more work will be needed. Rethink the scaling of the emissions, improve the description of the scientific basis behind the scaling as it is currently somewhat lacking. Furthermore make the improved CMAQab version the focus of the manuscript. In the current state it is mentioned in only a few sentences, raising the question why it is included in the first place, except to somewhat improve the diurnal variation, at which CMAQab is currently doing a poor job.

---

## Author Response (AR2)

**Author Response to Reviewer 1 – Modeling the Diurnal Variability of Agricultural Ammonia in Bakersfield, California during the CalNex Campaign**

*We would like to sincerely thank both reviewers and the editor for their time and contribution to reviewing this paper. We have responded below, with reviewer comments in bold, and author response in italics.*

**Suggestions for revision from Reviewer 1:**
**The recent revision has greatly improved the manuscript, and in my opinion could be improved to be ready for publication after a final set of revisions. The requested revisions mostly cover the results & discussion as to my opinion some conclusions are made to lightly and/or are not completely supported by the results the authors show. A few comments/edits are minor details and will only take a few minutes to correct. A few others are more in depth and touch the basis of the manuscript. The authors do not need to completely agree to all statements but some elaboration will be needed to convince me and to my expectation most of the readers.**

**In depth comments/discussion:**
**Overall & page 11 line 12-20 about the scaling factor of the emissions; You describe that you directly scale emissions needed to match a ratio of measured to modelled concentrations. What you more or less assume in this case is that the ammonia emissions is linearly related to ammonia concentration.**
**Can you shortly discuss the scientific basis behind this assumption, and why not use a somewhat lower or constant factor?**

*Our reasoning for choosing the diurnal scaling emission was due to the fact that the modeled to measured ratio of $NH_x$ at the Bakersfield site suggested that the largest error was due to a missing factor with a diurnal pattern. While the CARB inventory did not include diurnally-varying $NH_x$ emissions, other emission inventories have included such a distribution (Zhu et al. 2015a, Bash et al. 2015), so we chose to hypothesize this as a solution. The total emissions for the day were kept as the original CARB emission inventory suggested, for each grid box of the model. We agree that lower or constant factors may also be representative, but were not explored in this paper, as our main goal was to address the apparent diurnal error from the surface measurements and how it affected model comparisons with the TES instrument; additional model simulations were beyond the scope of the project.*

*Our assumption of a linear relationship between ammonia emissions and ammonia concentrations was based on mass balance considerations. If wind speed, deposition, and PBL height are held constant a simple box model over the Bakersfield site would show a linear relationship between additional $NH_3$ emissions and the $NH_x$ concentration, since there is not enough sulfate to react with all the $NH_3$ (Seinfeld and Pandis, 2006). However, we believe that errors in other parameters (PBL height, deposition, etc.) affect modeled $NH_{3(g)}$ and $NH_x$ concentrations more strongly, and so we focused on these parameters in the rest of the paper. We added language describing the expected reaction of NHx in an ammonia rich region, Page 11 Lines 33-35 to Page 12 Lines 1-5:*

> *"The intense agricultural activities in the SJV generate large $NH_3$ emissions, with concentrations often exceeding 5 ppb as indicated in the ground measurements, making this an $NH_3$ rich region relative to the ambient sulfate concentrations. In this regime, since there is not enough sulfate to react with all the $NH_3$, a simple box model over the Bakersfield site, with wind speed, deposition, and PBL height variation held constant, would show a linear relationship between additional $NH_3$ emissions and the $NH_x$ concentration (Seinfeld and Pandis, 2006). Thus we expect errors in other parameters (PBL height, deposition, etc.) to*

*affect modeled $NH_{3(g)}$ and $NH_x$ concentrations to a greater degree, and we investigate these parameters below."*

**Page 10, line 10-14, you mention the effect of hourly varying emissions, and that the diurnal variation is missing. What about the monthly variation of the different sources?**

*Emissions in the CARB inventory do vary month to month, However, in order to study the monthly variation of emissions, we would need a much longer time period of data, as the CalNex campaign only ran from May to June of 2010. Thus this paper can only discuss varying emissions for that time frame. The summer time frame also allowed us to make the assumption that particulate formation played a minor role in the resulting $NH_{3(g)}$ concentrations, already discussed on the paper on Page 2 Lines 23-24. We have also added the following to Page 9 Lines 8-9:*

> *"While emissions do vary month-to-month, we do not explore seasonal variation in this study, since the measurement campaign only occurred during the months of May and June."*

**Overall & discussion of effects of transport; I do not fully agree with the explanation / conclusion that transport of ammonia does not seem to be a major factor. Although you cover the basis of wind direction there is a short discussion missing on the effects of wind speed. Towards the north-west the ratio between livestock/fertilizer applications is probably different compared to the local conditions at Bakersfield, CA. As my personal knowledge of the counties surrounding Bakersfield is non-existing I cannot couple the summary of the sources given in S2 to the concentrations measured at the Bakersfield site (when combining this to the wind speeds, and assuming a more or less constant wind direction. Would it be possible to add a figure in which you compare windplots of the model (a) with measured concentrations (b). Radially you can show the measured concentrations, coloring can show the wind speed (or vice versa). This will support the explanation that transport / horizontal misrepresentation of the emissions are not a major cause of the difference between model and measured concentrations.**

**To somewhat support my statement; you show that the model underestimates the RVMR to the north west compared to the satellite, while locally its basically the same, wind speeds increase to the end of the day compared to the overpass time of TES (13:30). Even if you would perfectly model the local emissions in the Bakersfield model cells, the misrepresentation to the north-west could possibly cause of bias shown in figure 3 & 9.**

*We have performed an extensive investigation in response to the reviewer's suggestion of looking into measured and modeled wind speed and direction versus $NH_3$ concentrations. Below are two plots showing the measured wind direction versus measured $NH_x$ (left) and CMAQ wind direction versus CMAQ $NH_x$ concentration (right) measured at the Bakersfield site that is now Figure 3 in the paper. We also include an additional Figure S5 in the supplement comparing modeled and measured wind speed.*

[Figure]

Figure 3. Wind rose of measured wind direction and NH$_3$ on the left, and modeled wind direction and NH$_x$ on the right where contours represent number of data points (hourly) per wind direction. Note the difference in scale, where values are in ppb.

[Figure]

Figure S5. Comparing measured and modeled wind speed (a) when coming from the southeast direction and (b) all other directions relative to the Bakersfield, CA site. Comparing measured and modeled $NH_3$ concentrations (c) when coming from the southeast direction and (d) all other directions. Colors describe the $CMAQ_{base}$ (solid blue), $CMAQ_{AB}$ (purple) and $CMAQ_B$ (green) modeled scenarios.

*In two locations in the paper we describe results from these figures on Page 11 Lines 2-18:*

*"During the nighttime there is a shift in wind direction to sources coming from the southeast. Cooling air from up in the eastern mountain ranges causes a mountain drainage effect into the southern valley area. This interaction of the mountain drainage combined with the typical low-level jet from the northern central valley creates a Fresno Eddy, as described in Michelson and Bao (2008). Figure 3 shows a wind rose for all points included in Figure 2, where measured wind direction and $NH_x$ concentrations are shown on the left, and modeled wind direction and $NH_x$ concentrations are shown on the right. It can be seen in Figure S5a that the nighttime wind measurements from the southeast generally have lower wind speeds ($< 4\ m\ s^{-1}$) and that the model does not capture the variation of these wind speeds very well. This may be due to some timing errors in that the model may not capture true winds within a 4 km grid box, which corresponds to about 1-2 hours in real time. In general, many of the higher modeled $NH_x$ concentrations appear to be occurring during nighttime when the model should have winds out of the southeast, thus there is large model bias for these points. As indicated by the performed HYSPLIT back-trajectories, and the description*

*of air flow in the southern valley, we assume that although the measurements indicate the immediate wind direction was out of the south-east, the air-mass's long-range transport still travelled over the Central Valley to accumulate emissions from that region before being recirculated by the Fresno Eddy to eventually come from the southeast. Thus, an overestimate of emissions in the Central Valley at night could*

5 *still contribute to a model overestimate of measurements coming out of the southeast, rather than this air mass having come from a cleaner source, east of the mountains. Additionally, for the remaining time periods and majority of measurements not out of the southeast at nighttime, the model does a better job at simulating wind speeds (Figure S3), with a large model bias in $NH_x$ concentrations remaining."*

10 *And additionally on Page 12 Lines 30-34:*

*"Furthermore, when we compare the modeled $NH_3$ to measured values coming from just the southeast at night (Figure S5), the model bias is reduced by about factor of 3.5. This suggests that although the model may not capture the immediate wind direction and wind speed at night, as explained above, because of the*

15 *long-range transport down the Central Valley that evolves into the Fresno Eddy, reducing emissions in this upwind region also reduces model bias for these points in time."*

*This is also supported by the HYSPLIT back trajectories we have run (Figure S2) and further demonstrated in the fact that when we do apply the diurnal emissions profile, we see a reduced model bias in all wind directions, with a larger reduction*

20 *in bias when the winds are out of the southeast (Figure S5c). Thus, even with the apparent errors in the modeled wind direction and speed at Bakersfield, the magnitude of emissions, assuming the linear response of concentrations to emissions in this $NH_3$ rich regime (> 5ppb), is too high during this time of day from upwind sources, which in this case we assume to be the Central Valley.*

25 **Table S2; Add some coordinates or relative position compared to Bakersfield. Or a small map showing the counties?**

*The position of Bakersfield, California is shown in Figure 1 as the red star, on the colored background of emissions, with the highest emissions located in the Central Valley. We have also added county lines to Figure 1 and feel this is sufficient to familiarize the reader with how emissions are distributed with respect to the ground site and the flight and satellite tracks.*

**Also what does Farming Operation mean? And what about livestock? Is this part of Farming Operation?**

*In the CARB emissions inventory there are 3 categories of $NH_3$ emissions, which includes 1) emissions from pesticide/fertilizer application, 2) farming operations, which in this inventory includes livestock agriculture in all forms*

35 *including handling of all excrement and 3) other $NH_3$ emissions that do not fall into either of the previous categories. This is already described in the paper on Page 12, Line 14-18.*

**Overall discussion of the statistics in Table 1 & 2; What I am missing is an in depth discussion of the statistics given in Table 1 and 2. What essentially is shown in Table 1 is that there is zero correlation between the model and the**

40 **measurements, i.e. any statements made on the bias will not convince anyone and I think it does not fully reflect the performance of the CMAQ model as even using CMAQab only gives a correlation of 0.05. A few lines on why these correlations are so low will improve the manuscript and re-establish faith in the CMAQ models capabilities to simulate NH3.**

**Table 2 & Section 4.2; At a first glance I would conclude that CMAQ base is better than CMAQab or CMAQb. Even though the diurnal variation is improved for the hours between 1 AM and 6 AM, the overall levels for the other 3⁄4 of the day are still too high or too low depending on the hour. Only a short discussion is given of CMAQB and CMAQab, page 12, line 24-31. You correctly point out that the emissions are now far too high for most of the day for both "improvements". The same is visible in Table 1 and somewhat less in table 3 as TES only gives a snapshot of the situation at 13:00.**

*We thank the reviewer for the comments regarding the statistics of the paper. We have chosen in the paper to mostly discuss model errors in terms of the mean bias and normalized mean bias, and to focus on those errors that can be addressed by adjusting the diurnal cycle of $NH_3$ emissions, as has been done for other inventories (e.g., Bash et al. 2015). We did this as the analysis of the Bakersfield surface observations showed a large diurnally-varying model error, which would be expected to significantly affect comparisons with datasets that did not cover the entire day, such as the satellite and aircraft observations. We believe that these errors needed to be addressed first before any further investigation into errors in the magnitude of the total emissions, day-to-day variation in emissions (possibly accounting for the poor observed correlation), or the vertical transport of $NH_x$ (possibly affecting the aircraft observations). While we plan to address these other errors in future work, we feel that our investigation into the possible sources of the diurnal errors is sufficiently complex that it needs to be summarized in its own manuscript.*

*However, we agree with the reviewer, especially in the ground and flight measurement discussion, that the small correlations ($r^2 < 0.1$) should be pointed out to the reader, thus we have done so by highlighting low $r^2$ values (less than 0.1) in italics, removed any wording suggesting we have made any 'improvements' to these correlations and, when possible, emphasized the low correlations to the reader. For example on Page 13 Lines 1-2 we have added:*

> *"However, we note that the correlation of all three-model scenarios remains very low ($r^2 < 0.06$), suggesting further model errors, such as the neglect of any day-to-day variation in $NH_3$ emissions in our simulations."*

*We feel, however, that since the original model runs ($CMAQ_{base}$) also have low correlations for these measurements, that the investigation of possible diurnal errors in ammonia surface fluxes is justified, and that this investigation and manuscript is useful to readers.*

**This brings me to a point to question the value of CMAQb and CMAQab to the manuscript. While the authors do a good job describing the possible causes of error in the base model, the new additions do not improve the model and are thus somewhat irrelevant in the current state besides showing that it is not correct to scale emissions following concentrations and that the current bi-directional schemes are far from perfect, things that have been shown before. I would like to put forward two possible approaches to make the manuscript ready for publication.**

**1. Change the manuscript to fully focus on the performance of CMAQbase & CMAQb. These two versions of the model have been described before in earlier publications and only a small addition to the result section, to better cover the CMAQ b results, will be needed to improve the manuscript enough for publication. Especially as possible causes of CMAQb are already mentioned in the discussion.**

**2. If the authors want to keep in the CMAQb model more work will be needed. Rethink the scaling of the emissions, improve the description of the scientific basis behind the scaling as it is currently somewhat lacking. Furthermore make the improved CMAQab version the focus of the manuscript. In the current state it is mentioned in only a few sentences, raising the question why it is included in the first place, except to somewhat improve the diurnal variation, at which CMAQab is currently doing a poor job.**

*We thank the reviewer for their very thoughtful and thorough discussion and suggestions. We have decided to follow approach 2 provided by the reviewer. In order to justify why the CMAQ$_{AB}$ modeled scenario was performed, we refer the reviewer to the new description of both our assumption of the linear relationship of emissions to concentrations, as well as the more thorough discussion on wind transport in the Bakersfield area, discussed in the above comment responses. We have also added discussion to Section 4.2 and 4.3 to focus more on the CMAQ$_{AB}$ modeling results in addition to adding CMAQ$_{AB}$ results to the new Figure 4, which allows for easier comparison to the original CMAQ$_{base}$ run. Figure 6 now includes a flight comparison using CMAQ$_{AB}$ results, instead of CMAQ$_B$, to enhance the discussion in Section 4.2 of aircraft comparisons, and vertically modeled NH$_x$. Finally, we have also included CMAQ$_{AB}$ results of CMAQ modeled RVMR in Figure 8, comparing to TES RVMR, to allow a more in-depth and visual comparison of how the CMAQ$_{AB}$ model performs. We feel that with these additions to the plots, discussion, and organization of sections that the paper is significantly improved, with thanks to the reviewer.*

**Minor Edits:**

**Table 1; Add a description of MB and MNB, also add this to the text. Add the number of observations?**

*We thank the reviewer for suggesting this description. We have added a description to the header of Table 1, and point the reader to this description in the text (Page 10, Line 10), and in Tables 2 and 3. The text reads: Mean Bias (MB) = mean modelled – measured, Mean Normalized Bias (MNB) = mean ([modelled – measured]/measured)*

**Discussion of Table 1; you show correlations of around 0.001 to 0.05. At this point statistics about slopes and MB become more or less irrelevant as you are applying fits to clouds of scatter. Maybe some explanation as to why the correlations are so low, at least mention it in the text. Correlations are almost not mentioned in the text, unless they are high ~0.7…, don't hide the fact that the model misrepresents the measured values even after the CMAQab additions.**

**Table 2; similarly to table 1, add a description of MB and MNB. Add the number of observations?**

**Table 3; add description of MB and MNB. Add the number of observations?**

*We have added discussion throughout the paper to emphasize the small correlations. For example, on Page 12 Line 8-10:*
> *"However there is still a clear model NHx overestimate overall (MB of 4.57 ppb and large MNB of 45.74 %, see Table 1), and the low correlation is not improved (r2 = 0.01)."*

**Figure 1; Colorbar label: Add an E to mission. Also change font to the same font of the colorbar ticks?**

*We thank the reviewer for pointing out the typo and have corrected this and the font.*

**Figure 2; If possible add Wind Direction, My explanation is added in the in depth comments.**

*We respond to the suggestion of wind direction in the comments above. We have added wind direction to Figure 4 (green line in bottom panel) as well as the addition of a windrose plot in the new Figure 3.*

**Figure 3; If possible make one big figure with 4 subplots using figure 3a b and 9 a b. This will make it easier for the reader to compare the old and new situations, Also add the Blue "observed" plot to 9b.**

*We strongly agree with the reviewer's comment here, and feel as though this new figure would create much easier comparisons, thus we have updated Figure 3 (which is now Figure 2) to include the additional plots of old Figure 9, and changed the text to reflect this, also included below (Page 15, Lines 10-19):*

> *"Model bias in both the night and daytime simulation of surface $NH_x$ is reduced in the $CMAQ_{AB}$ scenario. The total bias is significantly reduced from the factor 4.5 at night and 0.6 during the day compared to the $CMAQ_{base}$ scenario (Figure 4a). In $CMAQ_{AB}$, the model does well between the hours of 1:00 am and 6:00 am local time (Figure 4c), perhaps related to the lower emissions at this time of day when adjusted emissions are used assuming the linear relationship of emissions to concentrations. The remaining diurnal bias shows a relative model underestimate with a factor of ~0.6 at 10:00 local time and a relative model overestimate peaking at ~1.7 at 19:00 local time (Figure 4c), with average $CMAQ_{AB}$ modeled concentrations slightly higher in the afternoon and peaking around 19:00 (Figure 4d). It is interesting to note that the $CMAQ_{AB}$ bias relative to surface concentrations is small near the TES overpass time (e.g., crossing 0% between 13:00 and 14:00 local time, Figure 4c), which is consistent with the small bias seen in the comparison with the TES observations in Section 4.3."*

**Figure 4, Good figure, as for colors, maybe blue and red? Easier to distinguish the differences.**

*We recognize the reviewers comment, however feel as though changing the colors to red and blue may confuse it with the CMAQ model and measurement comparisons, thus we would like to keep these colors so as to make this a clear distinction.*

**Figure 5, Maybe also add 2010/06/16 CMAQ B for the top figure? Else remove the top subplot. What about CMAQ AB? Another possibility: Change it to 2 figures, one for 2010/06/16 and one for 2016/05/24 both with 3 subplots, CMAQ base, B and AB. Also show the figures in chronological order.**

*We recognize the reviewer's comments here, and have chosen to replace the $CMAQ_B$ (bottom panel) with a $CMAQ_{AB}$ comparison. We keep the same three panels otherwise to show the daytime ability of $CMAQ_{base}$ to capture 'hot spots' of ammonia (first panel), the nighttime ability of $CMAQ_{base}$, which seems to contain $NH_x$ in the bottom layer of the model (second panel), and finally the third panel, which shows the increase in vertically modeled ammonia in the $CMAQ_{AB}$ model. We feel that excluding any of these panels would be taking away from the in-depth comparison in the paper. We not that Table 2 still describes the relevant statistics for all other flights, with the low $r^2$ values (less than 0.40), are italicized to point out to the reader.*

**Figure 7, Add in CMAQ B, CMAQ AB for comparison.**

*We have added both $CMAQ_B$ and $CMAQ_{AB}$ to Figure 7 (now Figure 8) to compare to the CMAQbase run, shown below. The statistics of this figure remain in Table 3.*

[Figure]

Figure 8. Scatter plot of CMAQ$_{base}$ (blue), CMAQ$_B$ (green) and CMAQ$_{AB}$ (purple) versus TES NH$_3$ representative volume mixing ratios for TES special observation passes (TES$_{RVMR}$) during the CalNex campaign with statistics discussed in Table 3.

**Page 7, line 28; W is a weighting matrix, add a few words on how the matrix is calculated (possible effects following such a mapping).**

*We have clarified the description of the CMAQ representative volume mixing ratio (CMAQ$_{RVMR}$) to also include a description of the weighting matrix calculation as follows (Page 7, LN 24-30):*
"

$$x_{TES} = x_a + A(x_{CMAQ} - x_a) \qquad (1)$$

*and the RVMR is calculated as*

$$CMAQ_{RVMR} = W * x_{TES} \qquad (2)$$

*where xa is a vector of the TES a priori NH$_3$ concentrations, A is the averaging kernel matrix, xCMAQ is a vector of the interpolated CMAQ NH$_3$ values, and W is a weighting vector (Rodgers and Connor, 2003; Shephard et al., 2011). **W** basically weights each level according to the sensitivity of the TES instrument at*

*that level. It is calculated by summing the most significant rows of the averaging kernel at each level (see the appendix in Shephard et al., 2011 for details)."*

*References added to text:*

5  *Michelson, Sara A,, J.-W. Bao, 2008: Sensitivity of Low-Level Winds Simulated by the WRF Model in California's Central Valley to Uncertainties in the Large-Scale Forcing and Soil Initialization. J. Appl. Meteor. Clim., 47 ( 12 ) , 3131-3149, doi: 10.1175/2008JAMC1782.1.*

*Seinfeld, J. H. and Pandis, S. N.: Atmospheric Chemistry and Physics, Wiley-Interscience, New Jersey, 2006.*

**Author Response to Reviewer 2**

**Page 8, line 9. Define "RRTMG".**

15  *It now reads "Rapid Radiative Transfer Model code for General Circulation Model applications (RRTMG, Mlawer et al., 1997; Iacono et al., 2008)."*

**Page 8, line 6-8. Could you indicate the reasons why you need to use initial and horizontal boundary conditions from another model?**

*The WRF-ARW model is a limited-area model, thus requiring independent boundary-conditions data from a larger scale model. Additionally all Eulerian models require initialization input for the execution of simulations. For consistency, both the boundary and initial conditions are taken from the same model.*

25  **Page 11, line 23-24. I guess you mean "CMAQ_base" and "CMAQ_B"**

*Yes, we thank the reviewer for pointing this out and have made this correction.*

**Page 13, line 5. I don't think this conclusion can be made based on the figure. The locations of different sources of**
30  **NH3 are still not clear to me based on the figure.**

*The wording has been changed in an effort to be more clear. It now reads on Page 14, Lines 10-13,*
     *"Applying the TES operator to the CMAQ profiles and calculating the $CMAQ_{RVMR}$ allows us to compare the satellite and model datasets quantitatively, as described in Section 2.3. Surface $NH_3$ from the $CMAQ_{base}$*
35     *run (Figure 7a) and the TES $NH_3$ RVMR (Figure 7b) along a sample TES transect both identify the regions of large $NH_3$ sources and the spatial changes along the transect and demonstrate that the $CMAQ_{RVMR}$ is underestimated for the base run, particularly at higher $NH_3$ RVMRs."*

**Page 13, line 9. The r2 of 0.64 is not a well-correlated case to me. I don't think "CMAQ_base inventory does a good**
40  **job of capturing the spatial distribution of NH3 emissions near Bakersfield" based on the r2 of 0.64 and mean bias of**
**-2.67 ppbv. I suggest the authors either soften the words or remove this conclusion.**

*We have removed the strong words indicating the correlation is "good" however we feel as though the model can still qualitatively capture the regions of higher $NH_3$ emissions, as with the TES instrument. Thus we have changed the wording on Page 14, Lines 10-22, that describe the model evaluations with TES measurements.*

[revised manuscript text omitted]